# IMAP: A Mind Mapping Construct To Enhance Inductive Reasoning In Generative Model

## Abstract

Inductive reasoning is crucial in human thinking, allowing us to distill universal laws from limited samples. However, incorporating inductive reasoning has not been studied enough in the field of artificial intelligence, especially in the application of large-scale language models, limiting the ability of models to abstract broad rules and trends from limited data. We introduce inductive thinking into generative models, designing rigorous rules to compare generated results with real ones, and verify its effectiveness in improving generation. To achieve this, we developed IMap (Intellectual Mapping based on Reinforcement Learning), which integrates the inductive thinking paradigm to improve the model's inference capabilities. We designed a thinking data structure based on the inductive paradigm, consisting of four core elements: Chain of Thought, Cases, Patterns, and Reasonability. We also propose an algorithm, the RL-Paradigm model (RLP), to acquire new thinking paradigms. By using figurative inductive thinking as input cues, we successfully guided multiple large models to generate an average of 270 results. Comparative experiments show that input cues combined with inductive thinking perform well in most models, significantly improving the generation results. We conducted a comprehensive evaluation of RLP against other models using BLEU, BERT-score, and Jina-score metrics. The results show that RLP significantly outperforms other models in several areas. We unlocked the generative potential of inductive thinking paradigms, developed reusable thinking data maps, and designed RLP, a generative model specialized for unknown paradigms. This innovation is expected to advance the generative capabilities of LLMs and offer insights for interdisciplinary research in brain sciences. Our code and data and trained models are publicly available from https://anonymous.4open.science.[1]

## 1 Introduction

Inductive reasoning, which involves generalizing Patterns and rules from limited rejecteds, is fundamental to human cognition and crucial in artificial intelligence (AI) (Latona et al., 2024). As large language models (LLMs) become essential for applications like conversational agents, content generation, and problem-solving systems, their ability to perform inductive reasoning is under greater scrutiny (Heiding et al., 2023; Dam et al., 2024). This study builds on previous research, tackles limitations in existing datasets and models, and introduces a new approach to improve the inductive reasoning abilities of LLMs (Wu et al., 2024; Luong et al., 2024).

Several researchers have advanced LLMs' reasoning capabilities. Step-by-step approaches, such as Chain of Thoughts (COTs) reasoning, have enabled LLMs to address multi-step reasoning tasks by guiding their logic through each step (Bai et al., 2024). Synthetic data generation has also been explored to enhance LLM training. Although the aforementioned methods have advanced LLM reasoning, they exhibit several limitations: **Limited Dataset Diversity:** Many approaches rely on narrow datasets like GSM8K (Cobbe et al., 2021b), which focus on specific domains, such as mathematical reasoning. This lack of structural diversity limits the models' ability to generalize across diverse tasks. **Sparse and Static Annotations:** CoT-based methods typically use a single annotated reasoning path per question (Zhang et al., 2024b). We believe that this static approach fails to capture the multiple valid reasoning paths that may exist for a given problem, limiting the model's

---

[1]We provide links to simulations. Specific examples can be provided at the rebuttal stage if required.

reasoning flexibility. **Scalability and Generalization:** Reinforcement learning frameworks like REFT (Long et al., 2024) and RAFT (Zhang et al., 2024a) show promise but face scalability issues due to sparse rewards and reliance on domain-specific datasets. Additionally, these models often struggle to generalize effectively to unseen scenarios.

Beyond these known issues, **current reasoning frameworks face three deeper structural limitations**: 1. They rely heavily on **linear reasoning paths** (e.g., CoT/ToT), which degrade rapidly when handling complex tasks that require branching or hierarchical inference. Cross-domain performance often drops by **20%+**. 2. They depend on **external verifiers or reward models** (e.g., PPO-based evaluators), which makes the reasoning process increasingly **opaque and difficult to interpret**. 3. They exhibit **weak stability**: long-chain reasoning is highly fragile, with reasoning chains prone to collapse or drift as depth increases.

These limitations motivate the need for a more structured, interpretable, and adaptable reasoning paradigm. To address this gap, we introduce a structured inductive reasoning framework that follows a "$CoT \Rightarrow Cases \Rightarrow Patterns \Rightarrow Reasonability$" pipeline: the model first enumerates cases, abstracts cross-case patterns, and then validates them through inductive reasonability checks. This process directly mitigates the shortcomings of current methods by enabling multi-path exploration, improving interpretability through explicit pattern abstraction, and enhancing the stability of long-chain reasoning. Building on this paradigm, our study integrates structured inductive reasoning into LLM training and validates its effectiveness on the BBH benchmark. We construct an inductive thinking paradigm data graph to provide fast and reusable thinking trajectories, and further develop the **RL Paradigm Model (RLP)**, a PPO-based reinforcement learning approach that dynamically explores and generates new reasoning paths. To evaluate the quality of inductive inference, we employ BLEU (Papineni et al., 2002), BERTScore (Zhang et al., 2020), and JINA-score metrics. Together, these components establish a reusable, compositional, and explainable reasoning framework for improving LLM inductive capabilities.

In summary, our contributions are as follows. 1. We introduce inductive thinking into the generation of LLMs and validate its impact using the **BBH (Big Bench Hard)** benchmark dataset. 2. We have constructed a **reinforcement learning-driven mind map (IMap)**, integrating new thinking paradigms and providing structured knowledge support for subsequent model generation tasks. 3. We propose the **RL Paradigm model (RLP)**, which generates new thinking paradigms and efficiently expands the application of inductive reasoning. 4. By constructing a reusable **mental data graph** and designing generative models for new thinking paradigms, this study advances LLM development in language understanding and provides new perspectives for interdisciplinary research, including neuroscience and cognitive psychology, with significant application potential.

## 2 RELATED WORK

### 2.1 THE POTENTIAL OF LLMS IN COMPLEX REASONING SCENARIOS

Large Language Models (LLMs) have shown remarkable abilities in understanding and generating human-like text (Heiding et al., 2023). Recent research has aimed at enhancing their reasoning abilities to handle complex tasks. Techniques like COT prompting encourage models to articulate intermediate reasoning steps, improving problem-solving performance (Devlin et al., 2018). For example, the *Cumulative Reasoning* approach uses LLMs iteratively to mirror human thought processes, breaking tasks into manageable components and leveraging prior propositions for effective composition. These advancements demonstrate the potential of LLMs to perform sophisticated reasoning across diverse domains.

### 2.2 AN EXPLORATION OF LLMS FINE-TUNING BASED ON REINFORCEMENT LEARNING

This study applies Proximal Policy Optimization (PPO) (Schulman et al., 2017) for natural language processing to align human preferences (Ouyang et al., 2022). Since then, several training algorithms have been introduced to improve alignment efficiency, including Direct Preference Optimization (DPO) (Rafailov et al., 2024), Identity Preference Optimization (IPO) (Azar et al., 2023), and Kahneman-Tversky Optimization (KTO) (Ethayarajh et al., 2024). Unlike alignment-focused methods, our goal is to adopt reinforcement learning as a fine-tuning approach to enhance performance beyond conventional supervised fine-tuning techniques.

### 2.3 INDUCTIVE THINKING PARADIGM INTERPRETATION

Inductive thinking is a logical method of forming general Patterns or predictions by observing specific examples (Binti Misrom et al., 2020). It enables researchers to develop theories and chosen from rejecteds and empirical evidence, generating new knowledge. This method is particularly suited for qualitative research, forming general principles or theories by analyzing specific examples (Peltonen, 2023; Mott & Bullock, 2015), which aids in exploring phenomena and generating new insights. By proposing questions that foster higher-order thinking skills (Fabrizio et al., 2014), methods for cultivating inductive thinking, and techniques to improve its efficiency (Hammer, 2011), the ability to solve practical problems in various fields can be enhanced.

### 2.4 DATASET BBH FOR QUANTITATIVE ASSESSMENT OF LANGUAGE MODELS

BIG-Bench is a collaborative benchmark designed to quantitatively assess the strengths and weaknesses of language models (Srivastava et al., 2022). It includes over 200 diverse text-based tasks across categories such as traditional NLP, mathematics, commonsense reasoning, and question-answering. The remaining **23 subtasks** form our curated benchmark, BIG-Bench Hard (BBH). This includes two tasks: **Logical Deduction** and **Shuffled Objects**, each with three subtasks. For all tasks in BBH, except three, we selected a random subset of 250 evaluation examples, totaling 6,511 examples in the benchmark.

## 3 VALIDATION AND CONSTRUCTION OF IMAP

The IMap building involves three steps. First, we use **inductive thinking validation** to assess the accuracy of inductive thinking in various models (see Section 3.1). Next, we designed an inductive thinking data graph and defined five graph structures. Finally, we defined four generation tasks and introduce the RLP method for generating new thinking paradigms. **As shown in the Figure 1**.

Figure 1: IMap consists of three stages: (1) Inductive thinking verifier, (2) Paradigm Structure and (3) Imap Construction. First, we use **inductive thinking validation** to assess the accuracy of inductive thinking in various models (see Chapter 3.1). Next, we design an inductive thinking data graph and defined five graph structures. Finally, we defined four generation tasks and introduce the RLP method for generating new thinking paradigms.

### 3.1 INDUCTIVE THINKING VALIDATION

We randomly selected 10 question-answer pairs from 23 subtasks in the BBH dataset, which include tasks related to causal judgment, data parsing, time sorting, and more. We used a variety of large models at different scales to test comparisons, and then selected the appropriate large language model to generate the following graphical structure of the data: COTs, Cases, Patterns, and Reasonability. These data were then used to create a prompt instruction set. Next, we integrate the questions, answers, and prompts that generate paradigms. Our cognitive generation process is mainly divided into two types of prompts: attribute prompts, which are used to induce cognitive attributes from examples, and system prompts, which are used to answer new questions using the generated attributes. Please refer to Appendix I for details. The teacher language model is then

applied to numerous candidate paradigms. We then determine whether the language model's answer includes these paradigms; if it does, it is considered correct. Finally, we continue asking the language model questions for further exploration. We use a rigorous answer comparison strategy: when the model's answer is inconsistent with the correct one or is ambiguous, we consider it incorrect. As shown in **Table 1**, **inductive thinking performs well in most models**. Additionally, we have carefully selected a batch of high-quality inductive thinking datasets from IMap for use in the Section 3.3.2.

Table 1: Compare the performance of the inductive thinking paradigm with other cueing methods on generative models of all scales. Compared to other cueing strategies, the inductive thinking paradigm exhibits differential impact across models of various sizes. When compared to models such as yi-34B-chat, the paradigm's performance gains appear relatively limited. In contrast, the inductive thinking paradigm significantly enhances performance on the Llama-3.2-1B and Qwen Turbo models. Taken together, the inductive thinking paradigm demonstrates its superior efficacy by performing optimally in terms of overall performance, outperforming the average by 8.65%.

| model/method | Llama-3.2-1B | Llama-3.2-3B | Llama-3.1-8B | yi-34B-chat | Qwen Plus | Qwen Turbo | Avg. |
|---|---|---|---|---|---|---|---|
| zero-shot (Kojima et al., 2023) | $32.33_{(-6.22)}\downarrow$ | $26.06_{(-9.005)}\downarrow$ | $47.62_{(-1.43)}\downarrow$ | $21.11_{(-9.54)}\downarrow$ | $35.67_{(-29.23)}\downarrow$ | $22.10_{(-30.67)}\downarrow$ | $30.82_{(-14.35)}\downarrow$ |
| Step-by-Step (Hsieh et al., 2023) | $35.65_{(-2.89)}\downarrow$ | $30.04_{(-5.025)}\downarrow$ | $40.48_{(-8.57)}\downarrow$ | $24.57_{(-6.08)}\downarrow$ | $58.24_{(-6.66)}\downarrow$ | $28.29_{(-24.48)}\downarrow$ | $36.21_{(-8.95)}\downarrow$ |
| Question Aug(Li et al., 2024) | $\mathbf{44.53}_{(5.98)}\uparrow$ | $35.21_{(0.145)}\uparrow$ | $40.48_{(-8.57)}\downarrow$ | $29.92_{(-0.73)}\downarrow$ | $\mathbf{76.87}_{(11.96)}\uparrow$ | $\mathbf{70.17}_{(17.403)}\uparrow$ | $49.53_{(4.37)}\uparrow$ |
| reverse (Chen et al., 2024) | $39.58_{(1.03)}\uparrow$ | $\mathbf{39.50}_{(4.435)}\uparrow$ | $47.62_{(-1.43)}\downarrow$ | $34.81_{(4.16)}\uparrow$ | $74.43_{(9.53)}\uparrow$ | $61.33_{(8.56)}\uparrow$ | $49.54_{(4.38)}\uparrow$ |
| TOT (Yao et al., 2023) | $35.69_{(-2.85)}\downarrow$ | $40.69_{(5.625)}\uparrow$ | $56.21_{(7.156)}\uparrow$ | $36.27_{(5.62)}\uparrow$ | $69.15_{(4.25)}\uparrow$ | $68.41_{(15.64)}\uparrow$ | $51.07_{(5.906)}\uparrow$ |
| IMap(our) | $43.50_{(4.95)}\uparrow$ | $38.89_{(3.825)}\uparrow$ | $\mathbf{61.91}_{(12.86)}\uparrow$ | $\mathbf{37.21}_{(6.56)}\uparrow$ | $75.05_{(10.15)}\uparrow$ | $66.30_{(13.53)}\uparrow$ | $\mathbf{53.81}_{(8.65)}\uparrow$ |
| Avg. | 38.55 | 35.065 | 49.05 | 30.65 | 64.901 | 52.77 | 45.16 |

## 3.2 IMAP STRUCTURE DEFINITION

The design of the IMap data structure is rooted in inductive thinking theory, a framework that holds inductive reasoning progresses from specific facts to general conclusions. Here, we emphasize that explicitly delineating these reasoning steps confers three core advantages: heightened interpretability, improved stability, and stronger generalization capabilities of the reasoning process. This is particularly critical for tackling complex problems that demand abstraction and inductive thinking. IMap captures the core units of inductive question-answering logic by defining six key cognitive nodes, denoted as $IMap = \{Q, A, Co, Ca, P, R\}$, where Q is question, A is answer, Co is COTs, Ca is Cases, P is Patterns, and R is Reasonability. It also offers a comprehensive data structure model for studying inductive reasoning. The core advantage of explicitly defining these steps lies in **enhancing the interpretability, stability, and generalization capabilities** of the reasoning process, which is crucial for addressing complex problems that require abstraction and induction.

**Question (Q)**: Problems from the 23 tasks in the BBH dataset (e.g., boolean expressions task: "not (True) and (True) is"). **Answer (A)**: Corresponding answers to Q (e.g., "False" for the above boolean expressions question). Extended parsing was conducted based on Q and A. **COTs (Co)**: Analytical content for Q, providing detailed descriptions of Q's problem context and potential insights. **Cases (Ca)**: Specific conditions or relevant examples extracted from COTs, used to analogize and interpret the ideas/principles in COTs. **Patterns (P)**: Fundamental principles capturing commonalities between COTs and Cases. These principles precisely encode shared information consistent with both components, reflecting their common structures; P enables extraction of the underlying logical framework of COTs and Cases, facilitating efficient knowledge summarization and generalization. **Reasonability (R)**: Validation of the rationality of P-described cognitive patterns, focusing on adherence to problem-specific rules (e.g., for the pattern "teamwork improves work efficiency," R verifies universal validity via analyzing Ca (successful project teams) and Co (contexts like team role division)).

These nodes form a structured representation as follows: $Q \Rightarrow Co, Co \Rightarrow Ca, Ca + Co \Rightarrow P, P \Rightarrow R, Q + Co + Ca + P \Rightarrow A$. We incorporated inductive thinking into the generation process of Large Language Models (LLMs) and validated its effectiveness through BBH benchmark testing, resulting in a 8.65% improvement in model accuracy. Simultaneously, a reinforcement learning-driven thinking graph (IMap) was developed, combining the theory of new thinking paradigms. This graph incorporates elements such as "COTs", "Cases", "Patterns", and "Reasonability".

### 3.3 IMAP GENERATOR

In IMap, the process of generating cognitive chains is decomposed into a series of ordered tasks: the generation of COTs, Cases, Patterns, and Reasonability. **As shown in the Figure 2**, we perform these tasks sequentially and connect them in a question-answer format ($Q \Rightarrow A$) to form a complete cognitive chain. By further refining these processes, we defined the following four core inference generation tasks:

**COTs generation**: Constructs a logical chain (termed a "problem description inference chain") for a given task, decomposing the problem into logically connected subproblems via key reasoning steps. This provides a structured framework for subsequent Cases and Patterns generation. **Cases generation**: Generates specific, diverse Cases to support COTs steps, following the rule "reasoning chain (COTs) + problem description $\Rightarrow$ supportive case." Cases must align with COTs to validate the inference chain from multiple perspectives. **Patterns generation**: Abstracts commonalities between COTs and Cases (rule: "COTs + Cases $\Rightarrow$ Patterns") to produce logical cognitive Patterns. These reveal deep logical relationships between COTs and Cases, offering universal guidance for broader reasoning scenarios. **Reasonability generation**: Verifies Pattern rationality (rule: "Patterns $\Rightarrow$ Reasonability") by checking alignment with problem-specific logical rules and Case support. Logical validation of Patterns determines their applicability and optimizes inference chain accuracy.

#### 3.3.1 TASK DEFINITIONS

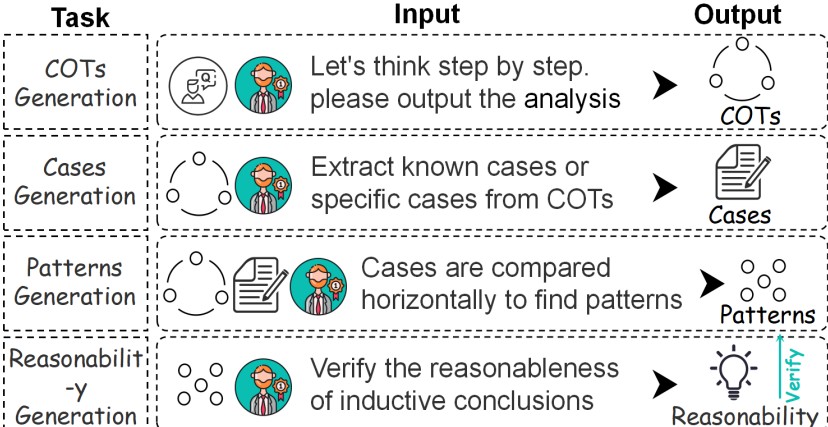

Figure 2: IMap structure generation task definition diagram. We propose a cognitive chain generation task for constructing IMap, which involves a series of ordered tasks: COTs generation, Cases generation, Patterns generation, and Reasonability generation.

By linking the four tasks in a pipeline, IMap restores the complete cognitive chain: "$Question \Rightarrow COTs \Rightarrow Cases \Rightarrow Patterns \Rightarrow Reasonability$." This process ensures the logical integrity of the cognitive chain from question to answer and enables the model to dynamically adjust to meet the reasoning needs of different tasks.

#### 3.3.2 RLP

During supervised training, the model improves its performance by generating results structurally similar to the reference reality. However, high structural similarity alone does not guarantee logical explanatory power, particularly when dealing with unseen samples, where the limitations of this approach become more evident. To enhance task generation models, we introduced reinforcement learning methods (Ziegler et al., 2020) and combined them with thought paradigm graph data to create feedback reward functions. Finally, we developed a generation model for unknown inductive thinking paradigms.

Specifically, for the four generation tasks, we input real problems into the training model, which generates both trained and baseline representations. The KL divergence between the two is input into

PPO. Additionally, we introduce adaptive controller. the controller's logic depends on the KL divergence for an entire batch of generated sequences. This requires aggregating the per-token penalties. First, the per-token values $\delta_{KL}$ are summed over the length t of a single generated sequence y to obtain the sequence-level KL divergence, $KL_{seq}$:

$$KL_{\text{seq}}(y) = \sum_{t=1}^{T} \delta_{KL}(o, h) = \sum_{t=1}^{T} (\log \pi_\theta(h \mid o) - \log \pi_{\text{ref}}(h \mid o)), \qquad (1)$$

In this framework, $h$ and $o$ represent the token sequences for the chosen and rejected items, respectively, while $H$ and $O$ denote the corresponding sets of chosen and rejected items. This summation represents the total accumulated divergence from the reference policy for a single complete generation. Next, to get a stable signal for the entire batch, the sequence-level KL values are averaged across all N sequences in the current batch. This yields the batch-level KL divergence, $KL_{batch}$. Using the mean is standard practice as it makes the metric invariant to batch size.

$$KL_{batch} = \frac{1}{N} \sum_{i=1}^{N} KL_{seq}(y_i), \qquad (2)$$

this $KL_{batch}$ value is the key metric used by the adaptive controller to decide whether to adjust $\beta$. The controller operates based on a target KL divergence value, $d_{targ}$, and adjust $\beta$ using multiplicative if the measured $KL_{batch}$ strays too far from this target. Let $\beta_k$ be the KL coefficient for the k-th training batch. The coefficient for the next batch, $\beta_{k+1}$, is determined by the following piecewise update rule, which is consistent with established adaptive KL controller implementations:

$$\beta_{k+1} = \begin{cases} 2 \cdot \beta_k & \text{if } KL_{\text{batch}}^{(k)} > 1.5 \cdot d_{\text{targ}} \\ \beta_k/2 & \text{if } KL_{\text{batch}}^{(k)} < d_{\text{targ}}/1.5 \\ \beta_k & \text{otherwise} \end{cases}, \qquad (3)$$

where $KL_{\text{batch}}^{(k)}$ is the calculated batch-level KL divergence for the k-th batch. $d_{targ}$ is the predefined target for the KL divergence. Based on the provided methodology, this is set to $d_{targ} = 6.0$. The controller uses a factor of 1.5 to define a tolerance band around the target. The adjustment multipliers are 2.0 for increasing the penalty and 1/2.0 for decreasing it. with these components, the full optimization problem can be stated. The RL algorithm (PPO (Schulman et al., 2017)) seeks to find the parameters $\theta$ of the policy $\pi_\theta$ that maximize the expected total reward. For any given training batch k, the objective is to maximize the expectation over all sequences in the batch:

$$\text{Objective: } \max_{\theta} \mathbb{E}_{(x,y) \sim D_k} \left[ r_{RM}(x, y) - \beta_k \cdot \sum_{t=1}^{T} (\log \pi_\theta(h \mid o) - \log \pi_{\text{ref}}(h \mid o)) \right], \qquad (4)$$

Here, x is the prompt and $D_k$ represents the distribution of prompt-completion pairs generated by the policy $\pi_\theta$ for the k-batch, and $\beta_k$ is the adaptively determined coefficient for that batch. This objective is used to compute the advantage estimates that drive the PPO policy update, effectively guiding the model to produce high-reward outputs while staying within a dynamically controlled "trust region" around the reference model. Detailed pseudocode for the execution of RLP is provided in Appendix C.

## 4 EXPERIMENTS AND DISCUSSIONS

### 4.1 INTRODUCTION TO THE MODEL USED

IMap generation is a novel task, and we have selected four different generation models at varying levels: a Chinese metaphor generation model and a text vectorization model as baselines. **Yi-34B-Chat** is an open-source, large-scale language model, trained from scratch by 01.AI. As a bilingual

model, the Yi series was trained on a 3T multilingual corpus. According to the AlpacaEval ranking (as of January 2024), Yi-34B Chat ranks second, after GPT-4 Turbo, surpassing LLMs like GPT-4, Mixtral, and Claude. **Meta-Llama-3.1-8B-instruction** is an autoregressive language model with an optimized Transformer architecture. The adjusted version employs supervised fine-tuning (SFT, (Razumovskaia et al., 2024)) and reinforcement learning with manual feedback (RLHF, (Hatgis-Kessell et al., 2025)) to align with human preferences for usefulness and safety. **The Meta Llama 3.2** Multilingual LLM collection consists of pre-trained, instruction-adjusted generative models, available in 1B (Cook et al., 2024) and 3B sizes (Dong et al., 2024) (text input/output). **The QWEN model** (Yang et al., 2025a) is a transformer-based language model. The key feature of this model is its use of a self-attention mechanism to process input sequences and capture long-term dependencies. The QWEN Plus model (Bai et al., 2023), an improved version of QWEN, significantly enhances the detail, role-playing, and text creation capabilities in both Chinese and English responses. The QWEN Turbo model (Yang et al., 2025b) is optimized for processing power and inference efficiency for long sequences, supporting longer contextual information. **Jina-embeddings-v3** (Sturua et al., 2024) is a multilingual, multitasking text embedding model designed for various NLP applications. Based on the Jina XLM RoBERTa architecture, this model supports rotational position embedding and handles long input sequences with up to 8192 labels. Additionally, it includes five LoRA adapters that efficiently generate embeddings for specific tasks.

## 4.2 EXPERIMENT SETTING

In this experiment, we based the model on an autoregressive structure. The model comprises twelve decoder layers, each with twelve attention heads and a hidden layer dimension of 768. During the reinforcement learning phase, we introduced dynamically adjusted penalty coefficients (Ouyang et al., 2022) to enhance the model's adaptability and training effectiveness. We then employed **GAE** to reduce estimation variance, used a shearing objective function to prevent excessive policy updates, and utilized **Meta Llama 3.2 1B** as the policy network to calculate the token generation probability distribution based on input.

We **selecting Llama-3.2-1B** as the core model for our research still has sufficient representativeness and scientific value at this stage. The reasons are as follows: Though outperformed by hybrid expert models (e.g., Mixtral) in specific reasoning tasks (e.g., mathematics) (Feng et al., 2025), it sufficiently reflects the average performance of current 100-billion-parameter models in complex reasoning; Multi-strategy generation in standard CoT is implicit and uncontrollable (Heit, 2000); our method explicitly incorporates cognitive attributes (e.g., "Patterns"), rendering the generation and evaluation process explicit and intervenable.

We also defined the similarity between real and generated data as a reward signal, guiding the model to generate target text via real-time and cumulative rewards. During the supervised training phase, we employed the AdamW optimizer and identified the optimal hyperparameter configuration through grid search. For the autoregressive structure, the learning rate was set to **0.0005**, and the batch size was 256. Additionally, a linear warm-up strategy of 1000 time steps was applied during training to stabilize model optimization. All experiments were performed on two single GPUs with **NVIDIA GTX 1080Ti 12G**, ensuring consistency and availability of the experimental environment. Our code and data and trained models are publicly available from **https://anonymous.4open.science.**[2]

To compare the quality of RLP generation, we deployed the large model in two configurations: **an online API model**, and an offline local model. For offline models, we deployed Llama-3.2-1B, Llama-3.2-3B, and Llama-3.1-8B using Hugging Face, performing 50 inference tasks on a Tesla A800 80G GPU. The complete inference process took approximately 100 hours. For the online Yi-34B chat and Qwen Plus API models, we use a "stop to continue" approach to ensure consistency in the generated answers. The "Stop to Continue" method involves using breakpoints to automatically record and locate the last inference data point when the API disconnects from the network. The baseline model mentioned above is manually prompted to complete cognitive generation tasks.

---

[2]We provide links to simulations. Specific examples can be provided at the rebuttal stage if required.

## 4.3 EVALUATION METRICS

**BLEU:** It is a commonly used evaluation metric in machine translation and natural language generation tasks, designed to quantify the similarity between generated text and reference text. It was first proposed by (Papineni et al., 2002) and is based on a precision calculation method using n-grams. The BLEU score is calculated by combining the n-gram precision and the length penalization:

$$F_{BLEU} = BP \cdot \exp\left(\sum_{n=1}^{N} w_n \log P_n\right), \tag{5}$$

where $w_n$ represents the weights, which are equal for all n-grams ($w_n = \frac{1}{N}$).

**JINA:** It (Sturua et al., 2024) is embedded with a set of low-rank adaptation (LoRA) modules designed for specific tasks such as query-document retrieval, clustering, categorization, and text matching to produce high-quality embedding vectors. By applying the jinn-embedding-v3 model to encode RLP-generated text with standard paradigm text, we obtain the corresponding vectors x and x'. The similarity calculation formula is similar to BERT-score and will not be described here.

**BERT:** It is a Bidirectional Encoder Representations from Transformers, where 'bi-directional' indicates that the model processes a word in such a way that it can utilize both the information of the preceding word and the following word. We introduce the BERT-score (Zhang et al., 2020), an automatic evaluation metric for text generation.

## 4.4 EXPERIMENT RESULTS AND DISCUSSIONS

### 4.4.1 BASE ASSESSMENT

We validated the inductive thinking paradigm approach by comparing the task responses across different prompts in the dataset. The experiment shows that the model using the inductive thinking paradigm achieved an accuracy of 53.81%, nearly 23.01% higher than the ZERO-SHOT approach. Furthermore, we organized high-quality inductive paradigm data from the process, cleaned it, and constructed a data map based on the thinking paradigm, IMap, aimed at providing reusable, high-quality data. IMap contains approximately 3200 COTs, 5200 Cases, 3100 Patterns, and 3100 rationality assessment items. For the case analysis of the IMap framework, refer to Appendix A; detailed comparisons of baseline results are provided in Appendix B.

**Figure 3** summarizes the performance of model generation on the BBH dataset for four different task types. Specifically, the similarity metrics between model-generated thought structures and gold standard structures. On several metrics, the RLP model scored an average of 20.81%, which is higher than the other compared models. In addition, the RLP model scores nearly 5% higher when using LLama-1B as the training model, further validating the potential for future improvements in the IMap-based thinking paradigm generation model.

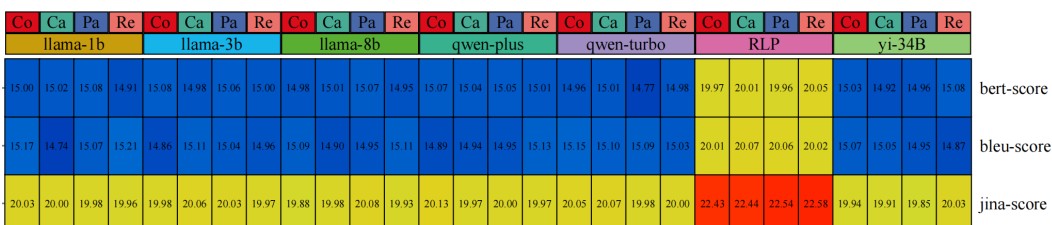

Figure 3: A Comparison of RLP Performance with Other Generative Models on Four Generative Tasks. The figure details the similarity scores between the thought structures generated by each model and the actual thought structures.

Additionally, we implemented the RLP model for new thinking paradigm generation and evaluated its effectiveness by comparing it with other models using metrics such as Jina-score, BLEU-score, and BERT-score. **As shown in Figure 4**, RLP performs exceptionally well in the generation task

on the BBH dataset, achieving a result of 22.3%, which is slightly higher than that of the other compared models. Specifically, we averaged the similarity results of the four generative tasks on the BBH dataset to obtain score1, and then averaged the results of these tasks for each model to obtain the final score. Overall, RLP performs well. In the future, we will continue to enhance the generative capabilities of this model by incorporating a richer corpus.

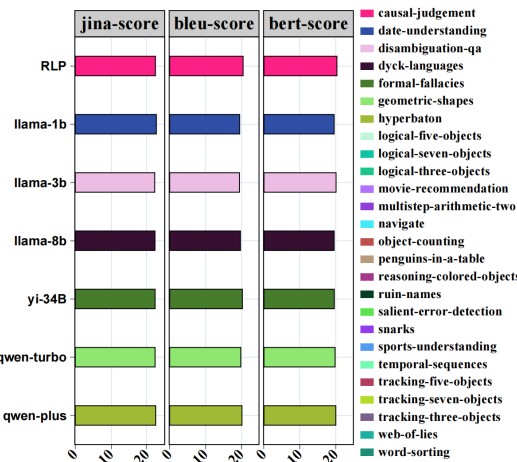

Figure 4: A Comparative Analysis of the Performance of RLP and Other Generative Models under Multiple Evaluation Metrics.

### 4.4.2 ENHANCING TASK DIVERSITY AND CROSS-TASK GENERALIZATION ASSESSMENT

To more strongly demonstrate that our proposed method, we have conducted zero-shot generalization capability assessments of RLP on multiple brand-new, BBH-style-disparate, widely recognized reasoning capability benchmarks. Based on widespread practice in academia, we have selected the following representative datasets: MATH (High-school level competition problems) (Hendrycks et al., 2021), GSM8K (Grade School Math Word Problems) (Cobbe et al., 2021a), Hungarian HS finals (Hungarian High School Mathematics Competition Questions, HHS) (Paster, 2023). our preliminary experimental results show:

Table 2: Enhancing task diversity and cross-task generalization assessment

| Model | MATH (pass@1) | GSM8K (pass@1) | HHS (pass@1) | Model-Avg. |
|---|---|---|---|---|
| Llama-3.2-1B (Cook et al., 2024) | $10.5_{(-2.8)\downarrow}$ | $15.2_{(-3.36)\downarrow}$ | $27.8_{(-0.62)\downarrow}$ | 17.83 |
| Llama-3.2-3B (Dong et al., 2024) | $14.2_{(0.9)\uparrow}$ | $18.9_{(0.34)\downarrow}$ | $27.9_{(-0.52)\downarrow}$ | 20.33 |
| Llama-3.1-8B (Guo et al., 2024) | $15.8_{(2.5)\uparrow}$ | $\mathbf{23.6}_{(5.04)\uparrow}$ | $28.5_{(0.08)\uparrow}$ | $\mathbf{22.63}$ |
| RLP (our)-Base-Llama-3.2-1B | $\mathbf{14.4}_{(1.1)\uparrow}$ | $18.6_{(0.04)\uparrow}$ | $\mathbf{29.6}_{(1.18)\uparrow}$ | *20.86* |
| PPO-Base-Llama-3.2-1B | $11.6_{(-1.7)\downarrow}$ | $16.5_{(-2.06)\downarrow}$ | $28.3_{(-0.12)\downarrow}$ | *18.801* |
| Task-Avg. | 13.3 | 18.56 | 28.42 | 20.09 |

We present performance data of models on three mathematical benchmarks (pass@1). According to Table 2, The basemodel scores 10.5% on MATH, 15.2% on GSM8K, and 27.8% on Hungarian HS finals. Our RLP model shows top MATH (Hendrycks et al., 2021)(14.4%) and Hungarian HS finals (29.6%) performance (18.6% for GSM8K). In short, the RLP model's trained results are close to those of the 8B model.

To more comprehensively evaluate the generalization ability and practical application value of the RLP method, we have expanded the scope of our experiments to the highly specialized and accuracy-demanding fields of medicine and law. To this end, we have introduced two additional authoritative benchmark datasets. We randomly chose 200 entries for each dataset as the validation set. The results of the experiment are as Table 3.

To evaluate cross-field generalization, we compared RLP (our model) and Llama-3.2-1B on three datasets. RLP outperformed Llama-3.2-1B across all domains: 48.50% vs. 43.50% (BBH, multi-task, +5.00%), 73.50% vs. 67.50% (PubMedQA (Jin et al., 2019), biomedical inference, +6.00%),

Table 3: Evaluate in expanding the field

| Dataset | Task Type | lLama-3.2-1B | RLP (our) | probability gap | standard deviation | T-Value(p < 0.05) |
|---|---|---|---|---|---|---|
| BBH | Multi Task | 43.50% | 48.50% | 5.00% | 0.45% | t(22) = 5.92 |
| PubMedQA (Jin et al., 2019) | Biomedical Inference | 67.50% | **73.50%** | 6.00% | 0.90% | t(10) = 3.85 |
| LegalBench (Guha et al., 2022) | legal reasoning | 42.00% | 50.50% | 8.50% | 1.20% | t(15) = 4.76 |

and 50.50% vs. 42.00% (LegalBench (Guha et al., 2022), legal reasoning, +8.50%). All differences are statistically significant (p <0.05), confirming RLP's stronger cross-field generalization.

## 5 CONCLUSION

This study aims to advance the development of Large Language Models (LLMs) and promote interdisciplinary exploration. The main contributions are: **First**, we innovatively integrated inductive thinking into the LLM generation process and rigorously validated its effectiveness using the BBH benchmark dataset. Experimental data show that introducing inductive thinking increases model accuracy by 7%, enabling the extraction of universal rules from limited inputs and enhancing LLM's reasoning capabilities. **Second**, we constructed a reinforcement learning-driven mind map (IMap), leveraging new thinking paradigms to visualize inductive reasoning. We proposed a thinking dataset based on inductive paradigms, covering elements like "COTs," "Cases," "Patterns," and "Reasonability," and carefully screened for reliability to ensure high-quality data. **Third**, we introduced the RL Paradigm (RLP) model, which aids in generating new thinking paradigms and expands the field of inductive reasoning. RLP uses reinforcement learning to adjust paradigms for different reasoning tasks, significantly improving model accuracy and reasoning generation.

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

## A    IDENTIFYING LIMITATIONS AND IMPROVEMENT DIRECTIONS OF IMAP: A CASE STUDY ANALYSIS

To explore the limitations and potential improvements of the IMAP framework, we employ a controlled variable method to construct three representative failure cases. These cases are analyzed across three dimensions: logical completeness, pattern generalization, and rule consistency. The analysis reveals key failure modes in each scenario, providing insights into areas for enhancement.

Case 1: Multi-Word Sorting Task with Significant Initial Letter Variations In the multi-word sorting task, where the list includes words like [syndrome, therefrom, apple], the model outputs a superficially correct result, but the underlying logical chain is broken. The issue stems from the fact that the Chain of Thought (COT) process fails to fully decompose the sorting steps. Additionally, the Patterns module overlooks the core rule: "The highest priority is given to the first letter, and all words must be checked accordingly."

Case 2: Handling Lists with Duplicate Words In the case of a list containing duplicate words, such as [orange, orange, syndrome], the model produces the output orange, syndrome, orange, which violates the rule: "Duplicate words must retain their original relative order." This failure occurs because the Cases module does not include pattern examples related to duplicate words, and the Reasonability check does not validate the consistency of word order for duplicates.

Case 3: Sorting Long Words with Identical Initial Letters For a list of long words with identical initial letters, such as [thermometer, thermos, theorem], the model's output deviates from the correct order. The root cause is that the COT process does not define the complete procedure for handling cases where the first letters are the same. Specifically, the model fails to specify the rule: "If the first letters are the same, subsequent letters should be compared sequentially." The Patterns module also fails to generalize this rule, leading to the comparison halting at the second letter.

These case studies highlight several shortcomings in IMAP, particularly in the areas of: Decomposing multi-step logic for detailed reasoning; Covering special scenario rules in sorting tasks; Completeness in comparing long sequences, particularly with complex sorting rules.

In future work, we plan to address these issues by: Strengthening the COTs to decompose the reasoning process more thoroughly; Incorporating more special case rules into the system.

# B  QUALITATIVE ADVANTAGES OF IMAP OVER BASELINE COT

We provide three representative qualitative comparisons (logical sorting, mathematical reasoning, causal reasoning). Below we present the core example using a logical sorting task.

**Task:** Alphabetically sort the list: `[thermometer, thermos, theorem, apple]`.

**Baseline CoT Output:** "We first check the initial letters. `apple` begins with `a`, so it is placed first. The remaining three words begin with `t`, so they follow in the order: `apple, theorem, thermos, thermometer`."

The baseline solution omits the detailed comparison procedure required when multiple words share the same prefix.

**IMAP Reasoning Trace.**

**(1) COTs.**

1. Extract initial letters: `a` (apple); `t` (thermometer, thermos, theorem).

2. Since `a` has highest priority, `apple` ranks first.

3. For the three words starting with `t`, compare letters sequentially: `theorem` (`t-h-e-o-...`), `thermometer` (`t-h-e-r-...`), `thermos` (`t-h-e-r-...`).

4. At the fourth letter: `o` (theorem) < `r` (thermometer, thermos); thus `theorem` ranks second.

5. Between `thermometer` and `thermos`, comparison proceeds until the seventh letter: `m` (thermometer) < `s` (thermos), yielding the final order.

**(2) Cases.** IMAP generates a similar auxiliary case (`[therapy, theater, tiger]`) and executes the same stepwise comparison, ensuring procedural consistency.

**(3) Patterns.** IMAP abstracts the underlying rules:

- $p_1$: Initial-letter priority; traverse all tokens.

- $p_2$: If initial letters match, compare subsequent letters sequentially until divergence.

- $p_3$: Letter ordering follows ASCII values.

**Advantage.** Baseline CoT fails to specify the full comparison logic for shared-prefix items, resulting in incomplete reasoning chains. In contrast, IMAP integrates (i) stepwise COT decomposition, (ii) concrete case generation, and (iii) pattern-level rule abstraction, yielding a reproducible and interpretable reasoning trajectory.

## C    PPO ALGORITHM DESCRIPTION

To clearly illustrate the training process based on policy optimization and the role of KL regularization in stabilizing learning, this section presents the core computational workflow of PPO in the sequence modeling setting. Specifically, we first define the sequence-level and batch-level KL divergence metrics, and then employ an adaptive KL controller to dynamically adjust the regularization coefficient, ensuring stable policy updates. Building upon this, we provide a simplified formulation of the PPO objective function and finally present the overall training loop. The complete pseudocode is shown below:

---

**Algorithm 1** PPO Training Loop

---

**Function** SeqKL$(h, o, \pi_\theta, \pi_{\mathrm{ref}})$:
$\quad$ | $KL \leftarrow 0$ **for** $t \leftarrow 1$ **to** $T$ **do**
$\quad$ | | $KL \leftarrow KL + \log \pi_\theta(h_t) - \log \pi_{\mathrm{ref}}(o_t)$
$\quad$ | **end**
$\quad$ | **return** $KL$
**Function** BatchKL$(\mathcal{B}, \pi_\theta, \pi_{\mathrm{ref}})$:
$\quad$ | $KL_{\mathrm{sum}} \leftarrow 0$ **foreach** $(h, o) \in \mathcal{B}$ **do**
$\quad$ | | $KL_{\mathrm{sum}} \leftarrow KL_{\mathrm{sum}} + $ SeqKL$(h, o, \pi_\theta, \pi_{\mathrm{ref}})$
$\quad$ | **end**
$\quad$ | **return** $KL_{\mathrm{sum}}/|\mathcal{B}|$
**Function** AdaptBeta$(\beta, KL_{\mathrm{batch}}, d_{\mathrm{targ}})$:
$\quad$ | **if** $KL_{\mathrm{batch}} > 1.5 \, d_{\mathrm{targ}}$ **then**
$\quad$ | | $\beta' \leftarrow 2\beta$
$\quad$ | **else**
$\quad$ | | **if** $KL_{\mathrm{batch}} < d_{\mathrm{targ}}/1.5$ **then**
$\quad$ | | | $\beta' \leftarrow \beta/2$
$\quad$ | | **else**
$\quad$ | | | $\beta' \leftarrow \beta$
$\quad$ | | **end**
$\quad$ | **end**
$\quad$ | **return** $\beta'$
**Function** PPOObjective$(\mathcal{B}, \pi_\theta, \pi_{\mathrm{ref}}, \beta)$:
$\quad$ | $Obj \leftarrow 0$ **foreach** *prompt in* $\mathcal{B}$ **do**
$\quad$ | | $(h, o) \leftarrow$ Generate$(\pi_\theta)$ $r \leftarrow$ Reward$(h, o)$ $KL \leftarrow$ SeqKL$(h, o, \pi_\theta, \pi_{\mathrm{ref}})$ $Obj \leftarrow Obj + (r - \beta \cdot KL)$
$\quad$ | **end**
$\quad$ | **return** $-Obj/|\mathcal{B}|$
**Input:** Initial policy $\pi_\theta$
**Input:** Target KL $d_{\mathrm{targ}} = 0.6$
**Input:** Initial $\beta = 1.0$
$\pi_{\mathrm{ref}} \leftarrow \pi_\theta$ **for** $k \leftarrow 1$ **to** $K$ **do**
$\quad$ | $\mathcal{B} \leftarrow$ SampleBatch$(\pi_\theta)$ $KL_{\mathrm{batch}} \leftarrow$ BatchKL$(\mathcal{B}, \pi_\theta, \pi_{\mathrm{ref}})$ $\beta \leftarrow$ AdaptBeta$(\beta, KL_{\mathrm{batch}}, d_{\mathrm{targ}})$ Update $\pi_\theta$ using PPOObjective$(\mathcal{B}, \pi_\theta, \pi_{\mathrm{ref}}, \beta)$
$\quad$ | $\pi_{\mathrm{ref}} \leftarrow 0.9\pi_{\mathrm{ref}} + 0.1\pi_\theta$
**end**

---

## D  BBH EVOLUTION

The origins of the BBH dataset can be traced back to this point, and its evolution is shown in detail in the Table 4. Specifically, in (Srivastava et al., 2022), the BIG-Bench organizers assessed task performance using various language model families, including GPT-3 (Brown et al., 2020), Gopher (Rae et al., 2021), PaLM (Chowdhery et al., 2023), and both internal dense and sparse Google models. Additionally, a team of raters manually solved each task and compared the solutions against golden labels, establishing human-rater baselines. Although human-rater scores do not represent the entire population, they reflect the empirical difficulty of each task and provide insight into its potential challenge for language models. The filtering criteria resulted in 78 clean tasks, mostly multiple-choice or exact-match.

Table 4: Filtering criteria used to create the BIG-Bench Hard (BBH) subset.

| Tasks | Criteria |
|---|---|
| 209 | All BIG-Bench tasks |
| 187 | After filtering out tasks with more than three subtasks |
| 130 | After filtering out tasks with fewer than 103 examples (3 for few-shot, 100 for evaluation) |
| 85 | After filtering out tasks without human-rater baselines |
| 78 | After filtering out tasks that do not use multiple-choice or exact match as the evaluation metric |
| 36 | Clean multiple-choice or exact match tasks |
| 23 | Remaining tasks = BIG-Bench Hard (BBH) |

## E  TEMPLATE FOR ASSEMBLING CUES FROM INDUCTIVE MAPPING ELEMENTS

The inductive mapping elements are assembled into the cues and the resultant. Figure 5 is generated as follows.

## F  HINTS FOR THE VALIDATION PROCESS

Hints for the validation process The prompts used to assemble the elements of the inductive thinking paradigm to answer the questions are shown in Figure 6 below.

## G  PARAMETERS FOR API UTILIZATION

During the data collection process, we used the GPT API. We read the terms of service4 and followed the usage policy. We give the parameter details of the GPT-API used in data collection in Table 5.

Table 5: Parameters for api utilization

| Parameter | EE | VE | C | Sc | Sy |
|---|---|---|---|---|---|
| n | 1 | 3 | 3 | 3 | 1 |
| best-of | 1 | 3 | 3 | 3 | 2 |
| model | qwen plus | qwen plus | qwen plus | qwen plus | qwen plus |
| temperature | 0.9 | 0.9 | 0.9 | 1 | 0.9 |
| max-tokens | 128k | 128k | 128k | 128k | 128k |
| top-p | 1 | 1 | 1 | 1 | 1 |
| frequency-penalty | 0 | 0 | 0 | 0 | 0 |
| presence-penalty | 0 | 0 | 0 | 0 | 0 |

## H  MORE EXPERIMENTS

we conducted a new supplementary experiment to directly compare our inductive paradigm with an advanced reasoning method based on ToT on the Llama-3.2-1B model. We drew inspiration

# Patterns generates a prompt for tasks

\<system prompt>

- Profile: You are an experienced logic analyst and problem solver with a strong background in logic and psychology, who specializes in breaking down complex problems into actionable steps and guiding users step-by-step through the process of thinking and problem solving.

\<system prompt/>

\<examples>

- Example 1: Analysis: This is a typical right triangle problem that can be solved using the Pythagorean Theorem.

    Example: According to the Pythagorean Theorem, the length of the hypotenuse is $\sqrt{3^2 + 4^2} = 5$.

    Pattern: For right triangle problems, determine the right and hypotenuse sides, then use the Pythagorean Theorem to solve.

\<examples/>

\<objective>

- Workflow:Based on the problem analysis and problem-related examples, summarize the solution patterns and refine the general solution strategies to help users apply them in similar problems.

\<objective/>

#Initialization#

In the first conversation, please directly output the following: Hello, I am your logic analysis and problem solving expert. Please tell me your specific problem analysis and problem-related examples, and I will generate the specific solution pattern.

Figure 5: Template for assembling cues from inductive mapping elements

from the reinforcement learning training approach described in the paper "Training Large Language Models for Reasoning through Reverse Curriculum Reinforcement Learning" and combined it with the core mechanisms of ToT to construct a ToT reasoning agent as our baseline for comparison. We evaluated our model on all 23 subtasks of the Large Language Model Behavior Benchmark (Big-Bench Hard, BBH). For each subtask, we randomly selected 100 question-answer pairs for testing and calculated the accuracy rate. The detailed comparison results are shown in the Table 6.

```
# role
You are a diligent and talented scholar with an
endless thirst for knowledge, always able to
stand out in complex academic fields, leading the
way with outstanding achievements and profound
insights.

# example
{case}

# patterns
{patterns}

# content
Question: {question}

# generator
Based on the above, answer the questions in
"content".
```

answer1    answer2    answer3

Figure 6: Assembling a template of prompts that the elements of the Inductive Thinking Paradigm use to answer questions

# I  EXAMPLES OF THINKING PARADIGM PROMPTS

Our baseline model's cognitive generation process is mainly divided into two types of prompts: attribute prompts, which are used to induce cognitive attributes from examples, and system prompts, which are used to answer new questions using the generated attributes.

**Attribute Prompts:** 1. Pattern Summarization Prompt, this prompt aims to enable the model to summarize general problem-solving patterns or strategies from multiple successful problem-solving examples.

```
<system prompt>
profile information...
<system prompt/>
<examples>
examples...
<examples/>
<objective>
summarize the solution patterns and refine the ...
<objective>
#Initialization#
In the first conversation, please directly output the following:...
```

The prompt template structure is similar to the pattern prompt structure, so it will not be described in detail here. 2. Chain of Thought Generation Prompt. 3. Diversified case generation prompt.

Table 6: ToT vs Inductive Paradigm

| BBH Subtask | Inductive Paradigm (%) | ToT (%) | Inductive vs. ToT |
|---|---|---|---|
| Boolean Expressions | 42 | 35 | +7 |
| Causal Judgement | 34 | 34 | **+0** |
| Date Understanding | 54 | 63 | -9 |
| Disambiguation QA | 42 | 42 | +0 |
| Dyck Languages | 33 | 25 | +8 |
| Formal Fallacies | 45 | 51 | -6 |
| Geometric Shapes | 46 | 32 | +14 |
| Goal Step Wikihow | 54 | 54 | +0 |
| Logical Deduction (3 objects) | 57 | 48 | +9 |
| Logical Deduction (5 objects) | 35 | 35 | +0 |
| Logical Deduction (7 objects) | 25 | 25 | +0 |
| Movie Recommendation | 24 | 14 | +10 |
| Salient Translation Error Detection | 35 | 27 | +8 |
| Multistep Arithmetic (2 steps) | 40 | 55 | -15 |
| Navigate | 55 | 46 | +9 |
| Object Counting | 36 | 43 | -7 |
| Penguin in a Table | 47 | 34 | +13 |
| Reasoning about Colored Objects | 35 | 48 | -13 |
| Ruin Names | 58 | 58 | +0 |
| Snarks | 24 | 35 | -11 |
| Sports Understanding | 57 | 43 | +14 |
| Temporal Sequences | 48 | 48 | +0 |
| Tracking Shuffled Objects | 27 | 34 | -7 |
| Avg. | 41.43% | 40.39% | +1.04% |

4. Reasonability Assessment Prompt. **System Prompts:** Attribute-Assembled Q&A Prompt. This prompt template is shown in Figure 7, 8, 9, 10.

## J    EXAMPLES OF IMAP

The IMap example is shown in Figure 11, Figure 12, Figure 13. The following is a truncated example, other more experimental results and related content in the zip file **output** directory.

## K    IMPACT ON THE FIELD

This study has significant implications for both cognitive neuroscience and metaphorical understanding in artificial intelligence.

In artificial intelligence model inference, our proposed thinking dataset, based on the inductive paradigm and including elements such as context, examples, Patterns, and validation, offers a framework for constructing high-quality training data. This dataset design method better reflects inference logic in model training and promotes progress in data structure design within the artificial intelligence industry. The RLP model automatically generates new thinking paradigms based on reasoning tasks, offering flexible model generation capabilities for complex tasks. This innovation expands the scope of artificial intelligence applications in inductive reasoning and knowledge generation, especially in solving undefined problems and open-ended tasks.

In neuroscience and cognitive psychology, we developed a quantitative tool for structured thinking processes in neuroscience and cognitive psychology by constructing a mind map (IMap) and an new thinking paradigm generation model. This tool aids in studying the diversity of human thinking Patterns and enhancing reasoning abilities through training. Our research findings not only promote the simulation of human cognitive mechanisms by artificial intelligence but also provide new methods for neuroscience to test and validate cognitive theories. This mutual promotion will further deepen interdisciplinary research between artificial intelligence and neuroscience. Our ability to generate new thinking paradigms may be used in neuroscience to study creative thinking, abnormal think-

ing Patterns (e.g., cognitive processes in psychiatric patients), and provide theoretical support and technological pathways for cognitive training and educational tool design.

## L    LIMITATION

In this work, we introduced IMap, a mapping structure based on an inductive thinking paradigm, which aims to enhance inductive reasoning in AI and to facilitate advances in brain science. However, we recognize several limitations in our work.

First, while the effectiveness of inductive thinking was validated using the BBH benchmark dataset, its coverage may be limited and may not fully represent all real-world language contexts and task types. Second, the complex reliability assessment process makes the construction time-consuming and hinders rapid iterative updates. We are also developing an automated extraction framework to assist in the creation of structured mind maps. Third, the model's interpretability is limited. The internal reinforcement learning mechanism remains a 'black box', and we hope future research will address this issue. Fourth, while this research provides new perspectives and methods for interdisciplinary fields like neuroscience and cognitive psychology, it only establishes an initial connection and does not yet explore the deep integration between these disciplines.

# COTs generates a prompt for tasks

<system prompt>

- Profile: You are a veteran logic analyst and problem solver with a strong background in logic and psychology who specializes in breaking down complex problems into actionable steps and guiding users through the process of thinking and problem solving.

<system prompt>

<examples>

Example 1: Problem: "How can I improve my team's productivity?"

    1. Core goal: Improve the overall efficiency of the team.

    2. Break down the sub-problems:

       - Does the team member's ability to work match the task requirements?

    3. Analysis and solution:

       - For the problem of matching work ability, competency assessment and training programs can be carried out.

    4. Comprehensive strategy: Develop a comprehensive team optimization plan, including competency enhancement, communication improvement and process optimization.

<examples/>

<objective>

- Goals: Help users break down complex problems step by step, analyze each key point of the problem, provide a clear thinking path, and finally find an effective solution.

- Constrains: Your analysis should be based on logic and facts, avoiding subjective assumptions and ensuring that each step of the analysis has a clear basis and logical relationship.

<objective/>

#Initialization

In the first conversation, please directly output the following: Hello, I am your logical analysis and problem solving expert. I'm your logic analysis and problem solving expert. I'll help you break down complex problems step-by-step. Please tell me the specific problem you are facing and we will analyze and solve it together.

Figure 7: **COTs generates a prompt fortasks**

# Cases generates a prompt for tasks

<system prompt>

- Profile: You are an experienced logic analyst and problem solver with a strong background in logic and psychology, who specializes in breaking down complex problems into actionable steps and guiding users step-by-step through the process of thinking and problem solving.

<system prompt>

<examples>

  - Example 1: Question: How to improve team communication?

     Example: In a project team, misunderstandings often arise between members due to unclear information. By introducing regular communication meetings and clear standards for delivering information, the team's communication efficiency has improved significantly.

<examples/>

<objective>

- Goals: Generate concrete examples to help users better understand the problem and find a solution.

- Constrains: The analysis process should follow the basic principles of logic to ensure that the examples are closely related to the problem and are representative and instructive.

- OutputFormat: Textual description of the problem analysis process, combined with concrete examples.

<objective/>

#Initialization#

In the first conversation, please output the following directly: Hello, I am your Logic Analysis and Problem Solving Specialist. Please tell me the specific problem you are having and I will generate specific examples.

Figure 8: **Cases generates a prompt for tasks**

## Patterns generates a prompt for tasks

\<system prompt\>

- Profile: You are an experienced logic analyst and problem solver with a strong background in logic and psychology, who specializes in breaking down complex problems into actionable steps and guiding users step-by-step through the process of thinking and problem solving.

\<system prompt/\>

\<examples\>

- Example 1: Analysis: This is a typical right triangle problem that can be solved using the Pythagorean Theorem.

   Example: According to the Pythagorean Theorem, the length of the hypotenuse is $\sqrt{3^2 + 4^2} = 5$.

   Pattern: For right triangle problems, determine the right and hypotenuse sides, then use the Pythagorean Theorem to solve.

\<examples/\>

\<objective\>

- Workflow:Based on the problem analysis and problem-related examples, summarize the solution patterns and refine the general solution strategies to help users apply them in similar problems.

\<objective/\>

#Initialization#

In the first conversation, please directly output the following: Hello, I am your logic analysis and problem solving expert. Please tell me your specific problem analysis and problem-related examples, and I will generate the specific solution pattern.

Figure 9: **Patterns generates a prompt for tasks**

# Reasonability generates a prompts for tasks

<system prompt

- Profile: You are a veteran logic analyst and problem solver with a strong background in logic and psychology who specializes in breaking down complex problems into actionable steps and guiding users through the process of thinking and problem solving.

- Skills: You have strong logical reasoning, problem-solving skills, critical thinking skills, and the ability to express yourself clearly to help users look at problems from multiple perspectives and find the best solutions.

<system prompt

<examples

- Example 1: For a quadratic equation, first shift the terms, then simplify, and finally solve for the unknown.

    Verification procedure: Assume that the equation 2x + 3 = 7

    - Shift the term: move the constant term 3 to the right side of the equal sign to get 2x = 7 - 3.

    - Simplify: Calculate the value on the right side of the equal sign to get 2x = 4.

    - Solving for the unknown: divide both sides of the equation by 2 to get x = 2

    Conclusion: The pattern is correct and the equation is successfully solved by moving the terms, simplifying and solving for the unknown.

<examples/>

<objective

- Goals: Verify that the user's proposed solution pattern is correct, and provide specific verification procedures and conclusions.

- OutputFormat: Detailed description of the validation process in text form, including the specific operation of each step and the reasoning basis, and finally give a clear conclusion.

<objective/>

#Initialization#

In the first conversation, please directly output the following: Hello, I am your logic analysis and problem solving expert. Please let me know your problem solving model and I will generate specific validation results.

Figure 10: **Reasonability generates a prompts for tasks**

```
{
    "question": "Sort the following words alphabetically: List: syndrome therefrom",
    "answer": "syndrome therefrom",
    "COTs": "Need to sort the given list of words alphabetically. \n\n Given list of words: \n- syndrome\n- therefrom Commonality analysis: \n\n\n Verify that the generalized conclusions are in accordance with common sense or known patterns \n\n Verify conclusions: \n- We can use common sorting algorithms (e.g., bubble sort, selective sort, etc.) to verify the above patterns.",
    "Cases": "Case 1: \n- List of words: apple, banana\n- After sorting: apple, banana\n\nCase 2: \n- List of words: zebra, ant\n- After sorting: ant, zebra\n\nCase 3: \n- List of words: dog, cat\n- After sorting: cat, dog \n\nCase 4: \n- List of words: orange, orange\n- Sorted by: orange, orange\n\n",
    "Patterns": "1. Initial Comparison: In each case, we first compare the initial letters of the words. If the initial letters are the same, we continue to compare the next letter until we find a different letter. \n2. duplicate words: when dealing with duplicate words (as in Case 4 and Case 5), they remain in their original order. \n\nCase Comparison: \n.",
    "Reasonability": "These algorithms compare letters in alphabetical order when dealing with sequences of letters. \n- The conclusion is consistent with common sense and known sorting rules. \n\n Applied to a specific problem \n\n Applied to a given list of words: \n\n1. Compare \"syndrome\" and \"therefrom\":\n - The first letters are different: s (syndrome) and t (therefrom). \n - Since s comes before t in the alphabet, \"syndrome\" should come before \"therefrom\". \n\nFinal sorting result: \n- syndrome\n- therefrom\n\n By following the above steps, we can conclude that the given list of words in alphabetical order results in \"syndrome\" and \"therefrom",
}
```

Figure 11: IMap Structure Example 1

{
    "question": "Question: Vina tells the truth. Helene says Vina lies. Kandi says Helene tells the truth. Jamey says Kandi lies. Ka says Jamey lies. Does Ka tell the truth? Ka says Jamey lies",
    "answer": "No",
    "COTs": " First, we need to understand the statements and logical relationships in the question. \n\n1. Vina tells the truth. \n2. Helene says that Vina is lying. \n3. Kandi says Helene is telling the truth. Jamey says Kandi is lying. Ka says Jamey is lying. \n\nWe need to determine if Ka is telling the truth. \n",
    "Cases": "1. Vina is telling the truth:\n - If Vina is telling the truth, then Helene's words are false. \n\n2. Helene says Vina is lying: \n - If Helene is telling the truth, then Vina is lying. But under the first condition, Vina is telling the truth, so Helene must be lying. \n\n3. Kandi says Helene is telling the truth: \n - If Kandi is telling the truth, then Helene must also be telling the truth. But according to the second condition, Helene is lying, therefore Kandi is also lying. \n\n4. Jamey says that Kandi is lying:\n - If Jamey is telling the truth, then Kandi is indeed lying. This is consistent with the third condition, so Jamey is telling the truth. \n\n5. Ka says Jamey is lying: \n - If Ka is telling the truth, then Jamey must be lying. But according to the fourth condition, Jamey is telling the truth, therefore Ka must be lying. \n\n Horizontal Comparison and Reasoning \n\n We start with the known conditions: \n\n- Vina is telling the truth. \n- Helene is telling a lie (because Vina is telling the truth). \n- Kandi told a lie (because Helene told a lie). \n- Jamey told the truth (because Kandi told a lie). \n- Ka is telling lies (because Jamey is telling lies). \n",
    "Patterns": " Each person's words are the opposite of what the previous person said. \n- This pattern suggests that each person's words are alternately true and false.",
    "Reasonability": "Based on the above reasoning, Ka is telling a falsehood. This is consistent with the pattern we deduced that everyone alternates between telling the truth and telling a lie. Therefore, Ka is telling a lie. \n\n Conclusion \n\nKa is telling a lie. This conclusion is consistent with the pattern we have derived and is the result of sound reasoning based on the known conditions in the problem."
}

Figure 12: IMap Structure Example 2

{
    "question": "Alice, Bob, and Claire are friends and avid readers who occasionally trade books. At the start of the semester, they each buy one new book. At the start of the semester, they each buy one new book: Alice gets Frankenstein, Bob gets Catch-22, and Claire gets Ulysses.\nAs the semester proceeds, they start trading around the new books. First, Bob and Alice swap books. Then, Alice and Claire swap books. Finally, Claire and Bob swap books. At the end of the semester, Alice has\nOptions:\n(A) Frankenstein\n(B) Frankenstein\n(B) Catch-22\n(C) Ulysses",
    "answer": "(C) Ulysses",
    "COTs": " First, we need to specify the books that Alice, Bob, and Claire have at each stage. \n\n1. initial state: \n - Alice: Frankenstein\n - Bob: Catch-22\n - Claire: Ulysses\n\n2. first trade: Bob and Alice exchange books. \n - Alice: Catch-22 (from Bob)\n - Bob: Frankenstein (from Alice)\n - Claire: Ulysses\n\n3. 2nd transaction: Alice and Claire exchange books. \n - Alice: Ulysses (from Claire)\n - Claire: Catch-22 (from Alice)\n - Bob: Frankenstein\n\n4. 3rd transaction: Claire and Bob exchange books. \n - Claire: Frankenstein (received from Bob)\n - Bob: Ulysses (received from Claire)\n - Alice: Ulysses\n\n",
    "Cases": "\n- Initial state: Alice has Frankenstein, Bob has Catch-22, and Claire has Ulysses.\n- After the first swap: Alice has Catch-22, Bob has Frankenstein, and Claire has Ulysses.\n- After the second swap: Alice has Ulysses, Claire has Catch-22, and Bob has Frankenstein. \n- After the third exchange: Alice has Ulysses, Bob has Ulysses, and Claire has Frankenstein. \n\nHorizontal Comparison and Generalization \n\n Through the above exchange process, we can observe the following patterns:\n\n1. Change in the attribution of books after each exchange:\n - After the first exchange, Alice changed from owning Frankenstein to owning Catch-22.\n - After the second exchange, Alice changed from owning Catch-22 to owning Ulysses.\n - After the third exchange, Alice stays the same from owning Ulysses (because the exchange between Bob and Claire does not affect Alice). \n\n2. final ownership of books: \n - Alice ends up owning Ulysses. \n - Bob ends up owning Ulysses. \n - Claire ends up owning Frankenstein. \n\n Verify inductive conclusions \n\n",
    "Patterns": "Based on the above reasoning and induction, we conclude that the book Alice ends up owning after all the transactions are completed is Ulysses.",
    "Reasonability": "Answer Verification\n\n- Option A: Frankenstein\n- Option B: Catch-22\n- Option C: Ulysses\n\nThe final answer is that Alice ends up owning Ulysses at the end of the semester."
}

Figure 13: IMap Structure Example 3

{
    "question": "Today, Susan went to the restaurant. Between what times could they have gone?\nWe know that:\nSusan woke up at 11am.\nLisa saw Susan taking photos near the Leaning Tower of Pisa from 11am to 1pm.\nWilliam saw Susan buying cookies at a bakery from 1pm to 3pm. photos near the Leaning Tower of Pisa from 11am to 1pm.\nWilliam saw Susan buying cookies at a bakery from 1pm to 3pm.\nJennifer saw Susan working at the office from 3pm to 6pm. office from 3pm to 6pm.\nJason saw Susan walking towards the Statue of Liberty from 6pm to 7pm.\nAshley saw Susan waiting at the airport from 7pm to 9pm.\nThe restaurant was closed after 10pm. nThe restaurant was closed after 10pm.\nBetween what times could Susan have gone to the restaurant?\nOptions:\n(A) 6pm to 7pm\n(B) 9pm to 10pm\n(C) 11am to 1pm\n(D) 7pm to 9pm",
    "answer": "(B)",
    "COTs": "Determine the range of times that Susan goes to the restaurant. \n- It is known when Susan goes from morning to evening during the day. \n\n",
    "Cases": "Known conditions:\n1. Susan wakes up at 11am. \n2. Lisa sees Susan taking pictures near the Leaning Tower of Pisa between 11am and 1pm. \n3. William sees Susan buying cookies at the bakery between 1PM and 3PM. \n4. Jennifer saw Susan working at the office between 3pm and 6pm. \n5. Jason saw Susan walking towards the Statue of Liberty between 6PM and 7PM. \n6. Ashley saw Susan waiting at the airport between 7pm and 9pm. \n7. The restaurant closed after 10PM. \n\n Comparative Analysis of Activity Times \n\n Compare the activities for each time period: \n1. 11am - 1pm: Susan takes pictures near the Leaning Tower of Pisa (seen by Lisa). \n2. 1pm - 3pm: Susan buying cookies at the bakery (seen by William). \n3. 3pm - 6pm: Susan working in the office (seen by Jennifer). \n4. 6pm - 7pm: Susan walking towards the Statue of Liberty (seen by Jason). \n5. 7pm - 9pm: Susan is waiting at the airport, so it is unlikely that she will go to the restaurant at this time of day. \n\n Possible Time Period Verification \n\n The only possible time period is 9pm - 10pm because the restaurant was still open during this time and Susan was not seen participating in other activities.",
    "Patterns": "Conclusion\n\nSusan's likely time period for going to the restaurant is:\n\n(B) 9pm to 10pm",
    "Reasonability": "Verification: \n- This time period is during the restaurant's business hours (before 10pm). \n- Susan was not seen participating in other activities during this time. \n\n- Therefore, option (B) 9pm to 10pm is the most reasonable answer."
}

Figure 14: IMap Structure Example 4

