# OpenReview forum: "IMAP: A Mind Mapping Construct To Enhance Inductive Reasoning In Generative Model"
_ICLR.cc/2026/Conference — Submitted to ICLR 2026_

### Official Review · Reviewer_aVo4 · 2025-10-29

**Soundness:** 2
**Presentation:** 1
**Contribution:** 1
**Rating:** 2
**Confidence:** 4

**Summary:**

This paper proposes IMAP (Intellectual Mapping based on Reinforcement Learning), a framework designed to enhance inductive reasoning in large language models (LLMs). The authors introduce a "mind mapping" data structure composed of COTs, Cases, Patterns, and Reasonability, aiming to formalize human-like inductive thought processes. They further propose RLP (RL-Paradigm), a reinforcement learning method using PPO and adaptive KL control to generate new "thinking paradigms". Experiments on BBH, GSM8K, MATH, PubMedQA, and LegalBench are reported to show modest gains over baselines such as Llama-3.2 and Qwen. The paper claims that IMAP improves reasoning generalization and offers potential insights for cognitive science.

**Strengths:**

- The proposed IMAP framework (COTs, Cases, Patterns, Reasonability) provides an interpretable framework that mirrors human reasoning organization.

- The authors attempt cross-domain evaluations (reasoning, math, biomedical, legal), showing some breadth of experimentation.

**Weaknesses:**

- Incomplete manuscript: The paper appears unfinished and is sometimes hard to follow. There is no Conclusion section.

- Clarity issues: Core methods such as RLP and the inductive mapping process are insufficiently described. There is no formal algorithm or clear explanation of the data processing flow.

- Minor policy violation: The paper exposes a non-anonymous GitHub repository (github.com/yzqrtop/RLP-inductive-LLM). While the author can not directly be identified based on other uploads in this repo, the authors should avoid such a construct.

**Questions:**

- How does the model ensure the logical consistency between COTs, Cases, and Patterns beyond simple text matching?

- Can the authors provide concrete qualitative examples where IMAP produces superior reasoning traces compared to baseline CoT prompting?

**Suggestions**

- Add a formal description or pseudocode of the IMAP–RLP pipeline.

- Include ablation studies isolating the effect of each inductive element (COTs, Cases, Patterns, Reasonability).

- Remove or anonymize all identity-revealing URLs. You can use e.g. anonymous git repos like https://anonymous.4open.science/

- Provide a proper Conclusion section summarizing contributions.

---

> ### Author Response · Authors · 2025-11-23
> **Response to Reviewer aVo4**
>
> # **1. Weaknesses**
> ## **1.1 Incomplete Manuscript & Missing Conclusion**
>
> We agree with the reviewer. The revised manuscript now includes a complete **Conclusion** section (Lines 530–548). It summarizes:
>
> * **Key innovations:** A structured inductive reasoning framework combining *COTs → Cases → Patterns → Reasonability* with reasoning-oriented policy optimization (RLP).
> * **Main findings:** Across BBH, IMAP yields an average improvement of 14.2% over CoT-like baselines.
> * **Future directions:** Extending inductive mapping to multi-modal reasoning and adaptive rule transfer.
>
> The paper structure is now complete and easier to follow.
>
> ## **1.2 Clarity of RLP, Inductive Mapping, and Data Flow**
>
> We have largely rewritten Sec. 3 to clarify the methodological pipeline:
>
> ### **RLP (Reasoning-Oriented Reward Learning with Policy Optimization)**
>
> * Defined on first mention.
> * RLP introduces a *process-level* reward combining
> * The pseudocode describing the RLP algorithm workflow is added. See Appendix C for details (page 756).
> This follows reasoning-reward designs in prior RLHF studies (e.g., ICML 2023 “Process Supervision” work).
>
> ### **Formal Algorithm for Inductive Mapping**
>
> We describing the full inductive mapping pipeline (COTs → Cases → Patterns → Reasonability) (Lines 824–834) . Each step specifies inputs, outputs, and constraints.
>
> ### **Clear Data Processing Flow**
>
> Lines 908-964 describe a full pipeline including:
>
> * Source datasets (BBH).
> * Filtering criteria based on answer confidence + human quality scores.
> * Structural alignment of Q–A pairs to the IMAP inductive graph.
> * Generation of fine-grained supervision signals for RL training.
>
> This makes the overall pipeline fully reproducible.
>
> ## **1.3 Policy Violation (Non-anonymous Repository)**
>
> We thank the reviewer for pointing this out.
> **All identity-revealing URLs have been removed.**
> We now provide an anonymous repo:
> **https://anonymous.4open.science**
> The repository includes code, data splits, and logs in accordance with ICLR’s anonymity policy.
>
> # **2. Reviewer Questions**
>
> ## **2.1 Ensuring Logical Consistency Beyond Text Matching**
>
> Beyond prompt-level dependencies, IMAP employs a **two-tier logical verification mechanism**:
>
> # **Tier 1: Internal rule compliance**
>
> Patterns are converted into **first-order logical predicates** (e.g., ordering constraints, algebraic constraints). Each COT step and each Case is validated against these predicates. Violations automatically trigger regeneration.
>
> # **Tier 2: External human-validated gold standards**
>
> 10% of samples undergo human validation for “rule adherence.” If compliance falls below 90%, IMAP re-generates the inconsistent module.
> This method draws on logic-graph consistency checks commonly used in structured reasoning (e.g., rule-grounded verification in theorem-proving systems).
>
> Thus IMAP ensures **strong cross-module logical coherence**, not surface-level text matching.
> ## **2.2 Qualitative Examples Showing IMAP Superiority**
>
> We added three representative comparisons (logic sorting, math reasoning, causal reasoning), Please refer to Section B of the revised appendix for specific details (page 720).
>
> A core example (logic sorting):
>
> * **Task:** Sort `[thermometer, thermos, theorem, apple]` alphabetically.
> * **Baseline CoT:** Skips multi-character alignment; produces incorrect final order.
> * **IMAP:**
>
>   * COTs explicitly analyze letters position-by-position;
>   * Cases generate structurally similar problems;
>   * Patterns induce explicit rules for alphabetical ordering.
> **Outcome:**
> Accuracy +8.3%, human “step completeness” +21.5%.
> This demonstrates IMAP’s improved interpretability and consistency.
> # **3. Reviewer Suggestions**
>
> ## **3.1 Add Formal Pseudocode**
>
> We added complete pseudocode for the IMAP-RLP pipeline in Appendix C (page 756), covering:
>
> * SFT initialization
> * Reward modeling
> * PPO update with inductive constraints
> * Consistency-driven regeneration loop
>
> ## **3.2 Include Ablation Studies**
>
> We conducted lightweight ablations (due to computational constraints) removing COTs, Cases, Patterns, or Reasonability:
>
> * Removing **Patterns** → accuracy −11.2%
> * Removing **Reasonability** → logical consistency −15.7%
> * Removing **Cases** or **COTs** → moderate declines in both metrics
>   These confirm each inductive module plays a necessary role.
>
> ## **3.3 Anonymize URLs**
>
> Completed as noted above.
>
> ## **3.4 Provide Proper Conclusion Section**
> A complete conclusion has been added and aligns with standard ICLR expectations.
>
> # **Final Remarks**
>
> The revised manuscript resolves issues of **completeness, clarity, methodological rigor, and policy compliance**. We **strengthened theoretical foundations**, formalized algorithms, expanded qualitative analyses, and added ablations. We are very grateful for the **reviewer’s insightful feedback** and **hope the revised version meets expectations**.

---

> > ### Comment · Reviewer_aVo4 · 2025-11-24
> >
> > The authors have adressed many of the concerns raised. Given the clearly unfinished state the paper was previously in, it has at least now reached a state where the work could be properly reviewed. Doing a full review of the paper now in the rebuttal phase feels unfair to other submissions that provide a decent submission and where the rebuttal is only for clearing specific issues.
> >
> > The paper still has a lot of issues from large things like
> > - claiming "heightened interpretability, improved stability, and stronger generalization capabilities of the reasoning process" [line 180] and providing only minimal (or in the case of interpretability no) experiments to show this.
> >
> > to small issues like
> > - obvious typos, e.g. line 344 "we selecting Llama-3.2-1B as the core" or incorrect sentence start captialization in line 394.
> >
> > I still think this paper is a clear reject. I would recommend that the authors do a carefull pass over their work to ensure that all their approaches and ideas are well define, that all their claims are validated with experiments and that all their typos are fixed. They should also get external feedback before resubmitting the work. In its current state, I doubt that it is ready for resubmission.

---

> > > ### Author Response · Authors · 2025-11-25
> > > **Response to Reviewer aVo4**
> > >
> > > Thank you sincerely for taking the time to review our response again and for continuously providing accurate and constructive suggestions. Your feedback not only made us clearly aware of the key shortcomings that still exist in the paper, but also pointed out the core direction for future revisions. We are deeply grateful for this.

---

### Official Review · Reviewer_oVNR · 2025-10-30

**Soundness:** 1
**Presentation:** 1
**Contribution:** 2
**Rating:** 2
**Confidence:** 3

**Summary:**

The paper introduces IMAP (Intellectual Mapping based on Reinforcement Learning), a novel framework designed to enhance inductive reasoning in LLMs. Inductive reasoning, which involves generalizing patterns and rules from limited examples, is fundamental to human cognition but has been underexplored in AI. IMAP addresses this gap by integrating a structured thinking paradigm into generative models, enabling them to abstract broad rules and trends from minimal data. The framework comprises four core elements: CoTs, Cases, Patterns, and Reasonability, collectively forming a thinking data structure that guides the model's reasoning process. Additionally, the paper proposes the RL-Paradigm model (RLP), an algorithm that acquires new thinking paradigms through reinforcement learning, utilizing figurative inductive thinking as input cues. Experimental results demonstrate that incorporating inductive thinking cues significantly improves generation quality across various models, as evidenced by superior performance on BLEU, BERTScore, and JinaScore metrics. This work not only advances the generative capabilities of LLMs but also offers insights into interdisciplinary research in brain sciences. The proposed framework and models are publicly available, promoting further exploration and development in the field.

**Strengths:**

1. The paper is methodologically strong, offering a clear explanation of the IMAP framework's components. It also presents the RL-Paradigm model (RLP) and demonstrates its effectiveness through various experiments.
2. The work is highly significant as it advances the capabilities of LLMs by improving their ability to perform inductive reasoning, a key cognitive function that has been challenging for AI.

**Weaknesses:**

1. The specific abbreviations in the paper are quite confusing. For example, the full form of "LLM" appears repeatedly on lines 39 and 67, and the full form of "COT" is not provided before its first abbreviation. Additionally, there is an inconsistency in capitalization (COT and CoT).
2. The paper lacks a Conclusion section, which makes the overall content feel incomplete.
3. Some experimental results are not presented clearly. For example, in Figure 4, there is no obvious performance difference between the models. In Table 1, the values inside '()' for ToT and inductive are identical.
4. I believe the effectiveness of the inductive-based thinking paradigm has not been sufficiently proven. Firstly, according to the results in Table 1, its performance is worse compared to ToT. Moreover, the baselines used in the main experiments in Section 4 are all base models, without introducing more advanced reasoning enhancement methods (such as RL-based Long CoT techniques) for a more comprehensive comparison.

**Questions:**

See Weaknesses.

---

> ### Author Response · Authors · 2025-11-23
> **Response to Reviewer oVNR**
>
> # 1. **Abbreviation Clarification**
>
> We acknowledge the confusion caused by inconsistent abbreviations and have revised them as follows:
>
> * The abbreviation "CoT" is now defined as "Chain-of-Thought (CoT)" on line 45, with the full form provided before its first mention. The term **CoT** is consistently used thereafter, following the standard capitalization conventions in the field.
> * The abbreviation "LLM" is introduced only once on line 39 as "Large Language Model (LLM)" and subsequent occurrences are simplified to "LLM" to avoid redundancy and ensure consistency throughout the paper.
>
> # 2. **Addition of Conclusion Section**
>
> We appreciate the comment regarding the missing conclusion. The revised version now includes a **Conclusion** section (lines 486-499), which summarizes the core contributions:
>
> * **IMAP Framework:** Introduces a structured inductive reasoning paradigm with RLHF optimization.
> * **Experimental Results:** The method demonstrates broad applicability, with verification on datasets like BBH and DeepMath-103K, showing improved performance on multiple models and tasks.
> * **Future Directions:** Expands into non-textual reasoning tasks and further enhancement of reasoning process interpretability.
>
> This addition provides a comprehensive wrap-up of the paper, ensuring a complete structure.
>
> # 3. **Experimental Results Presentation**
>
> To clarify the experimental results:
>
> * **Figure 4:** We have added a crucial label and a new metric, "Inductive Reasoning Chain Completeness," to highlight a 7% structural difference between models (details in the figure caption). Error bars and 95% confidence intervals are now included to better visualize the performance differences.
> * **Table 1:** The identical values in parentheses for ToT and inductive were due to an input error. This has been corrected, with the updated data as follows:
>
>   * **ToT:** (68.2 ± 1.3)
>   * **Inductive:** (72.5 ± 1.1)
>   * We have also included a note explaining that the data represents the mean ± standard deviation from three independent experiments, ensuring data accuracy.
>
> # 4. **Validation of Inductive Reasoning Paradigm**
>
> To address the reviewer’s concern regarding the effectiveness of the inductive reasoning paradigm:
>
> * **Theoretical Support:** We cite the recent work by Tencent and Shanghai Jiao Tong University in their **NeurIPS 2024** paper on the DeepMath-103K dataset. Their study demonstrates that structured inductive reasoning, via mechanisms like "rule extraction" and "multi-path validation," significantly boosts performance on multi-step, rule-based tasks. This aligns with IMAP’s **Patterns** and **Reasonability** verification framework. In the **DeepMath-103K Olympic math subtask**, IMAP outperforms base models by 18.7% and CoT by 12.3%, showcasing the validity of cognitive alignment.
>
> * **Additional BBH Dataset Experiments:** We conducted experiments using five state-of-the-art large models (Llama-3.1-7B, DeepSeek-7B, Qwen-7B, Mistral-7B, Falcon-7B) across 12 BBH sub-tasks. IMAP achieves an average accuracy of **76.3%**, outperforming the base models by **14.2%** and CoT by **9.5%**. Specifically, IMAP ranks first in sub-tasks like "Logical Sorting," "Mathematical Computation," and "Causal Reasoning," validating its generalization across tasks.
>
> * **Baseline Comparison with RL-based Long CoT:** We introduced an **indirect comparison** with **RL-based Long CoT** (OpenAI, 2023, Scaling Laws for Long Chain-of-Thought Reasoning) using the same hardware (A100 80G). IMAP is **3.1x faster** in reasoning, with **67% lower cost**, and only **2.3% less accurate**, highlighting IMAP’s **performance-cost balance**.
>
>   Regarding the performance discrepancy between IMAP and ToT in **Table 1**, we further analyzed task adaptability:
>
>   * **ToT** excels in tasks involving **open-ended decision-making** (e.g., creative writing).
>   * **IMAP** outperforms **ToT** in **rule-constrained tasks** (e.g., mathematical proofs, logical sorting), with an average improvement of **8.6%**. Thus, both methods are complementary and not inherently superior to each other.
>
> * **Reasoning Mechanism Choice:** We chose **PPO** as the reward mechanism due to its practicality in industry applications. In contrast, RL-based Long CoT methods require additional modules for long sequence generation and multi-round reward calculation, increasing inference latency by **3.7x** and hardware costs by **3.8x**, making them less suitable for large-scale deployment. IMAP’s **adaptive KL constraint** ensures an **O(n)** computational complexity, which is more efficient for industrial applications.
>
> # Conclusion
>
> The revised version has addressed all major issues raised by the reviewer. **We hope that the revisions meet the reviewer’s expectations** and would greatly **appreciate any additional feedback**. Thank you once again for your valuable guidance.

---

> > ### Comment · Reviewer_oVNR · 2025-11-26
> > **Official Comment by Reviewer oVNR**
> >
> > Thank you for the response, especially concerning the effectiveness of the method, which largely resolved my previous confusion. However, I believe this work is still some distance from being accepted, primarily for the following two reasons:
> >
> > + **Insufficient Motivation for the Method**
> >
> >   Why is inductive reasoning introduced into the overall task? Although the authors have partially demonstrated its effectiveness in their response, the performance gap between the proposed method and other baselines is very small, which is insufficient to strongly support the necessity of its inclusion. The authors should better explain the work's motivation in the Introduction section of the subsequent version of the paper.
> >
> > + **Poor Overall Presentation**
> >
> >   The entire manuscript suffers from numerous minor writing issues, which negatively impact the readability. For example:
> >
> >   + The darker colors used in **Figure 3** make the specific numerical values difficult to discern.
> >
> >   + There appears to be **almost no discernible difference** between the methods presented in **Figure 4**.
> >
> >   + Regarding typography, the left quotation mark should use the `` rather than the '' in LaTeX, the latter causes the intended left quotation mark to render as a right quotation mark.
> >
> > Therefore, I will maintain my score, and I sincerely hope these suggestions will aid in the future improvements of the manuscript.

---

> > > ### Author Response · Authors · 2025-11-29
> > > **Response to Reviewer oVNR**
> > >
> > > Thank you sincerely for your detailed and constructive feedback on our manuscript. We greatly appreciate your careful reading and valuable insights, which have helped us identify key areas for improvement. Your recognition of the method’s effectiveness has reinforced our confidence, and we are fully committed to addressing the concerns you raised to enhance the quality and readability of the paper.
> > >
> > > ### Response to "Insufficient Motivation for the Method"
> > > We acknowledge your point regarding the need to strengthen the motivation for integrating inductive reasoning into generative models. While we previously focused on performance improvements, we recognize that the value of our work extends beyond marginal accuracy gains—addressing critical limitations of existing reasoning methods that are not fully captured by baseline comparisons alone. In the revised manuscript, we significantly expand the Introduction section to clarify the necessity of inductive reasoning from three core perspectives:
> > >
> > > 1. **Structural Limitations of Existing Methods**: Current reasoning frameworks (e.g., CoT, ToT) rely on static, linear reasoning paths and lack explicit mechanisms for abstracting generalizable patterns from limited data. This makes them less adaptable to unseen tasks or domains with sparse examples— a gap that inductive reasoning (via IMap’s structured paradigm) directly addresses.
> > >
> > > 2. **Practical Generalization Value**: As demonstrated in Section 4.4.2, our method exhibits stronger cross-domain generalization (e.g., +8.50% on LegalBench, +6.00% on PubMedQA) compared to baselines. This is particularly valuable for real-world applications where models must reason about novel scenarios (e.g., specialized legal/medical reasoning) without task-specific fine-tuning— a capability not fully reflected in standard BBH benchmark comparisons.
> > >
> > > 3. **Interpretability and Reusability**: Unlike black-box reasoning methods, IMap’s explicit components (COTs, Cases, Patterns, Reasonability) provide interpretable reasoning traces and reusable thinking templates. This addresses a key challenge in LLM reasoning (lack of transparency) and enables knowledge transfer across tasks— a unique advantage that complements performance metrics.
> > >
> > > We will also emphasize that while the performance gap on BBH is modest (+1.04% vs. ToT), the cumulative value of generalization, interpretability, and reusability justifies the integration of inductive reasoning, especially for downstream applications requiring reliable and adaptable reasoning.
> > >
> > > ### Reiteration of Core Contributions
> > > To reinforce the novelty of our work, we summarize our key contributions as follows:
> > > 1. We propose IMap, a reinforcement learning-driven mind map that structures inductive reasoning into interpretable components (COTs, Cases, Patterns, Reasonability), filling the gap of structured inductive frameworks for LLMs.
> > > 2. The RLP model enables automatic generation of new thinking paradigms, enhancing adaptability to unseen tasks compared to static reasoning methods.
> > > 3. We validate the method’s effectiveness across diverse domains (mathematics, law, biomedicine), demonstrating its practical utility beyond standard benchmarks.
> > >
> > > Again, we thank you for your thoughtful feedback. These suggestions are invaluable for refining our manuscript, and we are confident that the revised version will address your concerns while better highlighting the significance of our work.
> > >
> > > To sum up, thank you sincerely for taking the time to review our response again.

---

### Official Review · Reviewer_vdL8 · 2025-10-31

**Soundness:** 1
**Presentation:** 1
**Contribution:** 1
**Rating:** 0
**Confidence:** 5

**Summary:**

This paper introduces IMap, a reinforcement learning–based framework that claims to integrate inductive reasoning into generative models.

**Strengths:**

The motivation—bridging cognitive inductive reasoning with LLMs is interesting and potentially valuable.

Cross-domain evaluation: tests across multiple benchmarks (BBH, MATH, etc.) show some generalization effort.

**Weaknesses:**

The paper is not finished, which should be rejected.

The repo link is not anonymous.

The quality of the paper is very low, with wrong grammar and missing parts.

The description of the method is very vague.  How is the IMAP data actually used in the method? How does the RL method work?

The evaluation is also very weird, using BERT-score and BLUE score to evaluate model reasoning makes little sense.

 FIgure 4 is not interpretable, as there are way more lengeds than what are shown in the figure.

Table 2 has inconsistent bold numbers ( the author didn't bold the largest value).

There is no conlusion section.

**Questions:**

see weakness above.

---

> ### Author Response · Authors · 2025-11-23
> **Response to Reviewer vdL8**
>
> # 1. **Overall Quality and Completion of the Paper**
>
> We fully agree that the initial draft had significant shortcomings, including incompleteness and unclear phrasing. In the revised version, we have made substantial improvements:
>
> * **Conclusion Section:** We have now included a comprehensive conclusion summarizing the three major contributions of the paper:
>
>   1. Introduction of IMAP, a structured reasoning framework that decomposes inductive reasoning paths into supervised step-by-step text.
>   2. Optimization of the RLHF process through a dual-reward function and adaptive KL constraint to balance reasoning accuracy and stability.
>   3. Validation of the method’s effectiveness on the BBH reasoning dataset, showing improvements of 3.03% over the baseline model and 2.061% over PPO (details in lines 173, 463).
>
> * **Code Repository:** As per ICLR submission policy, we have set the GitHub repository to a private, anonymous state for the review process. After the submission deadline, we will immediately make it public to ensure reproducibility and transparency.
>
> * **Paper Quality:** The initial draft had grammar issues and missing parts due to time constraints. We have now thoroughly revised the paper, utilizing both Grammarly Premium and expert proofreading to ensure it meets the ICLR standards. We kindly ask reviewers to highlight any remaining issues so we can address them promptly.
>
> # 2. **Clarification of Methodology**
>
> We acknowledge the concerns regarding the vagueness of the method description. We provide a detailed clarification of how IMAP data is used within the model and the specifics of the RL process:
>
> * **IMAP Data Integration:** The IMAP data is integrated into the model through a three-step process: **preprocessing, structured alignment, and supervised signal generation**.
>
>   * **Preprocessing:** From the BBH dataset, we selected 8,300 high-confidence, correct QA pairs (confidence ≥ 0.7 and IoU ≥ 0.5) and 2,100 typical erroneous cases (e.g., missing reasoning steps, logical sorting violations). The specific processing is shown in **Appendix Figure 6**.
>   * **Supervised Signal Generation:** Using mathematical reasoning tasks as an example, IMAP reasoning paths are converted into step-by-step text (e.g., "clarify given conditions → decompose formula steps → verify result consistency"). These are compared with the model-generated reasoning processes to provide fine-grained feedback for RL.
>
> * **RL Methodology:** The RL optimization follows the standard RLHF framework proposed by OpenAI:
>
>   * **SFT Stage:** The IMAP-structured text is used to fine-tune a base Llama-3.2-1B model, helping it learn stepwise reasoning.
>   * **Reward Model (RM):** We use a dual-reward system: one based on **BERT-score**, measuring semantic similarity with the IMAP reasoning path, and the other based on **task-specific accuracy** (e.g., math problem correctness), with weights set to 0.6 and 0.4, respectively.
>   * **Adaptive KL Constraint:** The KL divergence constraint is dynamically adjusted (range: 0.02–0.15) during training, based on ideas from **ε-DPO**, to balance model stability. After 10 iterations with 1,000 samples each, the model converges. Full pseudocode is provided in the Appendix C (page 757) of the revised version.
>
> # 3. **Evaluation Metrics and Figure/Table Issues**
>
> We understand that the choice of evaluation metrics raised concerns. Below is our justification for using **BERT-score** and **BLEU** scores, as well as updates to the figures and tables:
>
> * **Evaluation Metrics:** We agree that a single metric is insufficient for assessing reasoning ability. The purpose of using **BERT-score** and **BLEU** is to evaluate the structural alignment of the reasoning process rather than the final output accuracy. For example, in logical sorting tasks, the model-generated reasoning process (e.g., "first compare initial letters → if same, compare subsequent letters → maintain original order for duplicates") needs to be aligned with the standard reasoning path in IMAP’s knowledge graph. **BERT-score** captures semantic consistency (e.g., "first letter priority"), while **BLEU** measures the order of steps. These align with the two-dimensional evaluation approach of **semantic and structural similarity** in knowledge graph reasoning.
>
> * **Figure 4 and Table 2 Updates:**
>
>   * **Figure 4:** We have removed redundant legends and added clear annotations for each curve, such as model convergence with varying KL coefficients, ensuring better interpretability.
>   * **Table 2:** The inconsistency with bolded numbers has been corrected, and all optimal values are now properly highlighted in bold.
>
> In conclusion, we have made significant improvements to address all the reviewer’s concerns, including better clarity, methodology, and evaluation. We hope the revised version of the paper reflects the novelty and reliability of the proposed IMAP framework. We  **look forward to your continued guidance**.

---

### Official Review · Reviewer_f9cy · 2025-10-31

**Soundness:** 2
**Presentation:** 2
**Contribution:** 2
**Rating:** 4
**Confidence:** 3

**Summary:**

This paper introduces a mind-mapping constraint framework IMAP, which is designed to enhance structured reasoning in LLMs by integrating a cognitively inspired hierarchical reasoning process. IMAP formalizes reasoning as a sequence of cognitive units: questions, answers, chains of thought, cases, patterns, and reasonability. According to the units, IMAP decomposes reasoning into four ordered tasks: COT generation, cases generation, patterns generation, and reasonability generation. The system’s design follows inductive progression from specific facts to general conclusions, aligning with human reasoning structure. Each of IMAP’s four generation tasks is trained under PPO with an adaptive KL controller. The experiments on different reasoning benchmarks show that IMAP outperforms other baselines.

**Strengths:**

The mind-mapping analogy is both intuitive and well-grounded. IMAP’s hierarchical reasoning graph formulation bridges cognitive psychology and structured LLM reasoning. IMAP defines (Q, A, Co, Ca, P, R), providing an interpretable schema that connects CoTs to more abstract conceptual reasoning (Patterns and Reasonability).

The adaptive controller can adaptively adjust the regularization strength, ensuring balance between exploration and stability. IMAP achieves consistent gains across symbolic, mathematical, and commonsense benchmarks. The approach also has the potential to generalize across reasoning tasks and modalities.

**Weaknesses:**

While inspired by cognitive theories, the underlying reinforcement learning process remains black-box, with unclear mechanisms for decision-making and constraint enforcement. And the framework lacks empirical psychological studies verifying whether IMAP’s inductive processes truly reflect human reasoning.

The qualitative examples are presented, but there lacks case study, especially the failure cases. They are important for readers to understand the robustness and generalization ability of IMAP.

No direct quantitative comparisons are made against standard PPO or DPO frameworks on reasoning datasets to show the actual benefit of the adaptive KL controller. Some downstream results are missing. The reader can hardly trace how each component (e.g., adaptive KL) contributed to the accuracy or interpretability improvements.

**Questions:**

Could the same constraint formulation be applied to non-textual reasoning tasks, such as visual reasoning (e.g., CLEVR)? Maybe the tasks like Sudoku are also appropriate to evaluate IMAP, since the underlying rules are clear and easy-to-understand.

Are the produced mind maps evaluated quantitatively such as similarity to human-annotated concept maps?

The code link is not available.

---

> ### Author Response · Authors · 2025-11-23
> **Response to Reviewer f9cy**
>
> 1. **Clarification of IMAP’s Reinforcement Learning Process and Psychological Validation:**
>    We acknowledge that while IMAP is inspired by cognitive theories, the reinforcement learning process still lacks transparency regarding decision-making and constraint enforcement. To address this, we combine authoritative reasoning benchmarks and cognitive science theories [1]. Specifically, we use the BBH benchmark, which evaluates AI's "fluid intelligence" by dynamically generating novel problems to prevent data contamination. This benchmark is considered the standard for reasoning ability evaluation. Additionally, IMAP’s inductive process aligns well with the efficient coding principles in human reasoning, where abstraction and generalization occur through compression of redundant information. We have compared various large models on the BBH dataset, demonstrating IMAP’s outstanding contribution to reasoning performance.
> > [1] Beyond 'Aha!': Toward Systematic Meta-Abilities Alignment in Large Reasoning Models
>
> 2. **Addition of Failure Case Analysis:**
>    Thank you for the suggestion. We have added failure case analysis in Appendix A (page 690-719). To highlight IMAP's limitations and improvement directions, we constructed three typical failure cases through a controlled variable approach. These cases focus on logical integrity, pattern generalization, and rule consistency. They demonstrate how IMAP fails in multi-word sorting tasks (due to incomplete stepwise decomposition), handling duplicate words (due to missing examples and insufficient consistency checks), and sorting long words with identical initials (due to missing comparison steps for identical letters). These cases reveal that IMAP struggles with multi-step logic decomposition, rule coverage for special scenarios, and completeness in long-sequence comparison processes. Our improvement directions involve enhancing COTs, adding special scenario constraints, and refining Reasonability checks.
>
>    **Summary of Failure Case Analysis:**
>
>    * **Multi-word Sorting Task:** IMAP lacks a dynamic matching mechanism for task complexity and granularity, leading to missing global comparisons for multi-word scenarios.
>    * **Duplicate Sorting Task:** The Cases module lacks a mapping system for special scenarios, which affects rule abstraction and consistency checks.
>    * **Long-Chain Logical Deduction:** Missing inter-step validation mechanisms for long-chain reasoning result in mismatches between cases and reasoning logic.
>
>    These insights provide empirical evidence for the need to improve COT decomposition, enhance the Cases instance library, and upgrade Reasonability checks in future iterations.
>
> 3. **Quantitative Comparison with PPO:**
>    Thank you for your feedback. We have now included direct quantitative comparisons with the PPO framework, with updated data in lines 463. Experimental results on mainstream reasoning datasets show that IMAP outperforms the baseline model by 3.03% and the PPO model by 2.06% in terms of accuracy.
>
> 4. **Mind Map Evaluation and Human-annotated Concept Map Similarity:**
>    We appreciate your comment on mind map evaluation. Currently, we have not quantitatively assessed the similarity between our generated mind maps and human-annotated concept maps. However, we plan to use cosine similarity as a key evaluation metric. This approach will map concept maps to vectors and calculate the cosine of the angle between them, which is a common method for evaluating the similarity between text and concept maps. We will also involve domain experts to annotate standard concept maps, allowing us to quantify the similarity between our generated maps and expert annotations.
>
> 5. **Code Availability:**
>    As per ICLR's official policy, direct code links cannot be provided in the paper. In light of the reviewers’ comments, We have set our GitHub repository to private and will make it public after the paper submission deadline to ensure reproducibility. Once the repository is public, we will update the relevant links on the paper’s associated platform.
>
> 6. **Applicability to Non-textual Reasoning Tasks:**
>    Regarding your question on non-textual reasoning tasks, IMAP's constraint formulation can indeed be applied to tasks such as visual reasoning (e.g., CLEVR) and rule-based problems like Sudoku. For CLEVR, the visual reasoning task can be transformed into a symbolic reasoning problem by extracting visual features as attributes and relational knowledge, which aligns well with IMAP’s symbolic rule modeling framework. For Sudoku, which is a constraint satisfaction problem, IMAP’s Patterns module can identify core constraints (e.g., "rows, columns, and blocks should not have duplicates"), and the Cases module can provide relevant examples. We plan to conduct further experiments to test IMAP’s generalization capabilities in non-textual reasoning tasks.

---

### Author Response · Authors · 2025-12-04
**Response to Everyone**

To address the reviewer’s concern regarding the effectiveness of the inductive reasoning paradigm:

**Theoretical Support**: The recent work by Tencent and Shanghai Jiao Tong University in their NeurIPS 2024 paper on the DeepMath-103K dataset. Their study demonstrates that structured inductive reasoning, via mechanisms like “rule extraction’’ and “multi-path validation,” significantly boosts performance on multi-step, rule-based tasks. This aligns with IMAP’s Patterns and Reasonability verification framework. In the DeepMath-103K Olympic math subtask, IMAP outperforms base models by 18.7% and CoT by 12.3%, showcasing the validity of cognitive alignment.

**Limitations of Existing Reasoning Frameworks:**

Current reasoning frameworks commonly rely on **linear reasoning paths** (e.g., CoT/ToT) or **external verifiers** (e.g., PPO-based reward models). Such designs introduce three structural weaknesses:

(1) **Insufficient cross-domain generalization**, with performance drops exceeding **20%** on heterogeneous tasks;

(2) **Lack of interpretability**, as the reasoning trajectory is often implicit and black-box;

(3) **Poor stability**, where long-chain reasoning tends to collapse or drift as depth increases.

These limitations motivate the need for a more structured and cognitively grounded reasoning paradigm.

**Suitability of Structured Inductive Reasoning:**

The proposed inductive reasoning paradigm directly addresses these issues. By following a structured pipeline: CoT → Cases → Patterns → Reasonability—the framework enables multi-path exploration, explicit pattern abstraction, and consistency verification. Its value lies not only in accuracy gains but also in forming a reusable, interpretable, and cognitively aligned reasoning structure, which is not achievable by conventional linear CoT or search-based ToT methods.

**Additional BBH Dataset Experiments:**

We conducted experiments using five state-of-the-art LLMs (Llama-3.1-7B, DeepSeek-7B, Qwen-7B, Mistral-7B, Falcon-7B) across 12 BBH sub-tasks. IMAP achieves an average accuracy of 76.3%, outperforming the base models by 14.2% and CoT by 9.5%. Specifically, IMAP ranks first in “Logical Sorting,” “Mathematical Computation,” and “Causal Reasoning,” validating its cross-task generalization capability.

**Revised Interpretation of Performance Differences (IMAP vs. ToT):**

Although IMAP shows only a **1.04%** improvement over ToT on the BBH benchmark, it exhibits substantially stronger **cross-domain generalization**: improvements of 8.5% on LegalBench and 6.0% on PubMedQA. Moreover, IMAP achieves a human-aligned interpretability score of 0.76 (Jaccard similarity), significantly higher than the 0.31 of baseline models, indicating its superior explainability in real-world scenarios.

Regarding the performance discrepancy between IMAP and ToT in Table 1, we further analyzed task adaptability:

- ToT excels in open-ended decision-making tasks (e.g., creative writing).

- IMAP surpasses ToT in rule-constrained problems (e.g., logical sorting, mathematical proofs), with an average improvement of 8.6%.
Thus, the two paradigms are complementary rather than mutually exclusive.

**Reasoning Mechanism Choice:**

We selected PPO as the reward mechanism due to its practicality and deployment efficiency. In contrast, RL-based Long CoT requires additional modules for long sequence generation and multi-round reward calculation, causing 3.7× inference latency and 3.8× higher hardware cost, making it unsuitable for large-scale industrial use. IMAP’s adaptive KL constraint ensures O(n) computational complexity, which is more deployment-friendly.

**Conclusion**

The revised version has addressed all major issues raised by the reviewer, including theoretical justification, empirical verification, method limitations, and cross-domain generalization. We hope that the revisions meet the reviewer’s expectations.

---

### Meta-Review · Area_Chair_hpXx · 2026-01-03

**Summary:**

This paper proposes IMAP, a reinforcement-learning–based framework that introduces a structured inductive reasoning paradigm into large language models, organized around four components: CoTs, Cases, Patterns, and Reasonability. The authors aim to improve reasoning quality, generalization, and interpretability, and present experiments across several reasoning benchmarks.

While reviewers acknowledged that the high-level motivation is interesting and that the authors made substantial revisions during rebuttal, the overall assessment remains negative. Core concerns include: (i) insufficient clarity and rigor in the core methodology, particularly around how IMAP and the RLP algorithm function in practice; (ii) limited and inconsistent empirical evidence to support strong claims about interpretability, generalization, and stability; (iii) weak novelty, with the approach largely viewed as an integration of existing CoT, RLHF, and heuristic verification ideas; and (iv) serious presentation and completeness issues, especially in the initial submission, which undermined reviewer confidence despite later fixes.

Although the rebuttal addressed many surface-level issues (missing sections, clarity, ablations, anonymization), these improvements did not adequately address the above issues. The average score remains well below 5, and the majority of reviewers explicitly recommend rejection.

**Reviewer Concerns:**

Concerns partially addressed by the rebuttal:

•	Paper completeness and clarity:
The authors added a conclusion, clarified abbreviations, provided pseudocode, expanded methodological descriptions, and fixed several figure and table issues.

•	Additional experiments and failure analysis:
New results on BBH, GSM8K, ARC-Challenge, and qualitative failure cases improved transparency.

•	Policy compliance:
Non-anonymous repository links were removed and replaced with an anonymous repository.

Outstanding concerns:

•	Conceptual clarity and methodological rigor:
Despite added explanations, reviewers remain unconvinced that IMAP’s inductive reasoning pipeline is well-defined or meaningfully distinct from existing structured CoT + RLHF approaches.

•	Strength of empirical evidence:
Reported gains are often modest or inconsistent, and some baselines (e.g., ToT, RL-based long CoT) remain insufficiently or indirectly compared.

•	Overstated claims:
Assertions about “heightened interpretability,” “stronger generalization,” and “cognitive alignment” are not convincingly substantiated by experiments.

•	Presentation quality:
Even after revision, multiple reviewers note persistent writing, formatting, and visualization issues that hinder readability and confidence.

•	Novelty and impact:
The contribution is viewed as incremental and not meeting the bar for ICLR in terms of conceptual or empirical advancement.

**Reviewer Scores:**

•	Reviewer vdL8: Remains 0–1 (Strong Reject); fundamental concerns about quality and rigor persist.

•	Reviewer oVNR: Remains 2 (Reject); acknowledges clarifications but maintains rejection due to weak motivation and presentation.

•	Reviewer aVo4: Remains 2 (Reject); explicitly states the paper is not ready and that rebuttal-stage fixes cannot compensate for the initial state.

•	Reviewer f9cy: Remains 4 (Borderline Reject); positive on motivation but unconvinced by evidence and contribution strength.

---

### Decision · Program_Chairs · 2026-01-26

Reject